# Panacea: Pareto Alignment via Preference Adaptation for LLMs

**Yifan Zhong**[1,2*], **Chengdong Ma**[1,2*], **Xiaoyuan Zhang**[3*], **Ziran Yang**[4], **Haojun Chen**[1,2]
**Qingfu Zhang**[3], **Siyuan Qi**[2], **Yaodong Yang**[1]

## Abstract

Current methods for large language model alignment typically use scalar human preference labels. However, this convention tends to oversimplify the multi-dimensional and heterogeneous nature of human preferences, leading to reduced expressivity and even misalignment. This paper presents Panacea, an innovative approach that reframes alignment as a multi-dimensional preference optimization problem. Panacea trains a single model capable of adapting online and Pareto-optimally to diverse sets of preferences without the need for further tuning. A major challenge here is using a low-dimensional preference vector to guide the model's behavior, despite it being governed by an overwhelmingly large number of parameters. To address this, Panacea is designed to use singular value decomposition (SVD)-based low-rank adaptation, which allows the preference vector to be simply injected online as singular values. Theoretically, we prove that Panacea recovers the entire Pareto front with common loss aggregation methods under mild conditions. Moreover, our experiments demonstrate, for the first time, the feasibility of aligning a single LLM to represent an exponentially vast spectrum of human preferences through various optimization methods. Our work marks a step forward in effectively and efficiently aligning models to diverse and intricate human preferences in a controllable and Pareto-optimal manner.

## 1 Introduction

AI alignment aims to ensure AI systems align with human intentions, and there has been notable progress in this area, especially for large language models (LLMs) [29, 12, 30, 2]. The prevailing approach for LLM alignment involves curating a dataset $\{(x, y_1, y_2, z)\}$, where each prompt $x$ is associated with a pair of responses $(y_1, y_2)$ and a scalar label $z \in \{0, 1\}$ that indicates if $y_1$ is a "better" response. These labels are typically generated based on detailed guidelines that encompass various criteria, reflecting multiple dimensions $i \in \{1, \cdots, m\}$ of human preferences (*e.g.*, helpfulness, harmlessness, conciseness, humor, formality). Pre-trained models are subsequently further optimized on this dataset using methods including reinforcement learning, supervised learning, or game-theoretical approaches [27, 42, 32, 6, 44, 4, 47, 40]. However, this *single-objective alignment* methodology may not fully capture the complexity of real-world scenarios for two reasons (Figure 1).

**First**, this method can lead to inconsistency and ambiguity in **data labels**. Human labelers assign scalar labels $z$ by *implicitly* evaluating responses across every dimension $i$ with *different preference weights* to $i$, and reaching a final judgment. These differences often result in conflicting labels, causing misalignment or learning failures (Appendix B), substantiated by the low average label agreement reported in [5]. **Second**, optimizing a single objective leads to only one **model** that attempts to fit the potentially conflicting labeling preferences, *i.e.*, the helpfulness-harmlessness dilemma. This

---

*Equal contribution. [1]Institute for Artificial Intelligence, Peking University. [2]State Key Laboratory of General Artificial Intelligence. [3]Department of Computer Science, City University of Hong Kong. [4]Yuanpei College, Peking University. Correspondence to: Yaodong Yang <yaodong.yang@pku.edu.cn>

38th Conference on Neural Information Processing Systems (NeurIPS 2024).

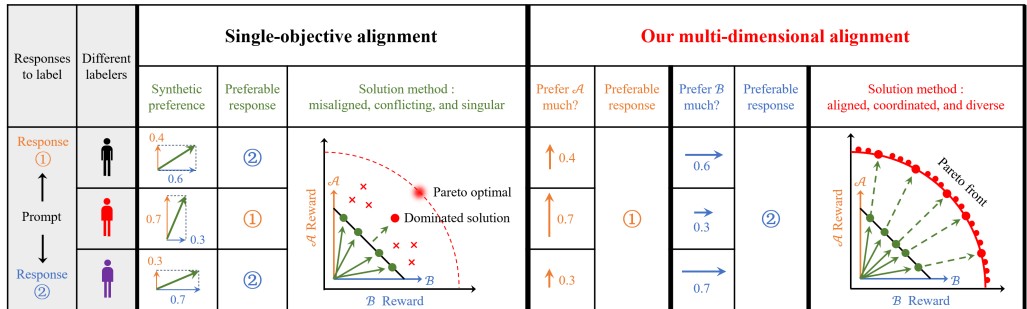

Figure 1: Comparison of the predominant single-objective alignment and our multi-dimensional alignment. For the two responses to a prompt, labelers agree on the preferable one in each preference dimension, but conflict when assigning a synthesized scalar label denoting which is "better". This arises due to the inherently different preference weights held by labelers, a common case in reality. Performing single-objective optimization on the potentially conflicting scalar-label dataset (left) could lead to a dominated solution and misalignment. By contrast, our method, Panacea, leverages multi-dimensional preference optimization (right) on the consistent multi-dimensional dataset and learns the entire Pareto front (PF), thereby aligning with diverse and complex human preferences.

single model may not cover the full spectrum of human preferences across all dimensions, thereby exacerbating biases against underrepresented groups and failing to meet diverse user needs.

To address these challenges, we formulate the alignment as a multi-dimensional preference optimization (MDPO) problem. By *explicitly* curating data for each dimension, we enhance data consistency and simplify the labeling process, thereby **overcoming the first limitation**.

Upon the obtained dataset, our goal is to concurrently optimize across all dimensions. However, this is often infeasible due to potential conflicts among preferences (*e.g.*, helpfulness *vs.* harmlessness in response to hazardous user requests). Therefore, we aim for Pareto-optimality [39], which means finding solutions where no preference dimension can be made better off without making another worse off. However, many Pareto-optimal solutions might exist. Instead of just learning one such solution, we focus on learning the entire set of Pareto-optimal solutions. To achieve this, we use a single model capable of recovering any Pareto-optimal solution by inputting the appropriate preference vector.

In this paper, we propose Panacea (**P**areto **a**lig**n**ment vi**a** preferen**ce a**daptation), a simple yet effective method that: 1) learns the entire Pareto-optimal solution set for all possible preferences with a single model, and 2) infers Pareto-optimal responses online by simply injecting any preference vector into the model. Our method, providing a comprehensive representation of human preferences, effectively caters to diverse user needs, thus **mitigating the second limitation** (Figure 1).

A key challenge lies in how to utilize a low-dimensional preference vector to control the model's behavior. Our core insight is that, similar to the crucial role of the preference vector in shaping the Pareto solution, singular values are pivotal in defining the model's fundamental behavior in a singular value decomposition (SVD)-based low-rank adaptation (LoRA)[22, 57]. To address the above challenge, we incorporate the preference vector into the singular values within each SVD-LoRA layer. We then scale it using a learnable factor to align with the magnitude of other singular values. The model is trained end-to-end using a joint objective function aggregated according to the preference vector. The flexibility of Panacea enables seamless compatibility with various preference optimization procedures, *e.g.*, supervised fine-tuning (SFT), reinforcement learning from human feedback (RLHF) [42], and direct preference optimization (DPO) [44], and diverse methods for loss aggregation, *e.g.*, linear scalarization (LS) [10][Section 4.7.5] and weighted Tchebycheff (Tche) [39][Section 3.4]. Through theoretical analysis, we confirm that Panacea can effectively capture the entire Pareto front (PF) under practical conditions. This finding provides a solid rationale for training a single Pareto set model to learn all Pareto optimal solutions across the entire preference space.

In our experiments, we assess the effectiveness and scalability of Panacea on several significant and challenging preference alignment problems with up to 10 dimensions, where the Pareto set

cardinality grows exponentially with the number of dimensions, considerably surpassing the scope of current research. Panacea consistently outperforms baseline methods, producing superior, uniformly distributed, and convex fronts in accordance with the theory. Quantitative metrics highlight its substantial advantages, demonstrating an order-of-magnitude improvement. Notably, Panacea exhibits no performance saturation even on the ten-dimensional problem, indicating its extensive potential. For the first time, we show the possibility of aligning a *single* model with *exponentially many* heterogeneous preferences, opening up a promising avenue for LLM alignment.

This paper makes three main contributions. **First**, we identify the fundamental limitations of the predominant scalar-label, single-objective alignment paradigm, and propose to reframe alignment as a multi-dimensional preference optimization problem. **Second**, we design Panacea, a simple yet effective method that learns one single model that can online and Pareto-optimally adapt to any set of preferences, without the need for further tuning. **Third**, we provide theoretical supports and empirical validations to demonstrate the Pareto optimality, scalability, efficiency, and simplicity of Panacea, thereby satisfying the urgent need for Pareto alignment to diverse human preferences.

## 2 Related Work

**Pareto Set Learning.** Different from previous classical multi-objective optimization (MOO) methods [59, 35, 38, 56] that use a finite set of solutions (referred to as "particles") to approximate the entire Pareto set, Pareto set learning (PSL) [41, 36, 58] aims to use a single model to recover the complete Pareto set/front. The advantage of PSL is that it can store an infinite number of Pareto solutions within a model. This allows users to specify their own preferences, and the model can dynamically output a particular Pareto solution in real-time according to those preferences. Typical applications of PSL includes multiobjective industrial design problems [58, 37], reinforcement learning [8, 54, 24], text-to-image generalization [33], and drug design [25, 61]. While there have been some studies on PSL involving deep neural networks, these models are considerably smaller compared to LLMs. Learning continuous policies that represent different trade-offs for LLMs remains unsolved.

**Multi-Dimensional Preference Optimization.** Existing research primarily treats AI alignment as a single-objective optimization problem with scalar labels [42, 55, 17, 44, 40, 47], often neglecting the complexity of diverse human preferences. We provide an in-depth analysis of this limitation in Appendix B, which is subsequently substantiated by MaxMin-RLHF's result of "impossibility of alignment" [13] **after Panacea first came out**. To address this crucial gap, one recent attempt is AlignDiff [19], which trains an attribute-conditioned diffusion model to conduct preference alignment planning in the RL settings. In the realm of LLMs, there are some contemporary works on this topic [60, 26, 18, 21, 51–53], where the most relevant one Rewarded Soups (RS) [45] adopts a multi-policy strategy. It learns a model for each preference dimension and interpolates their parameters linearly to generate a customized model. However, its simple design also constitutes its drawback. Since RS does not see any intermediate preference vectors during training, ensuring the optimality and alignment of the interpolated model poses a challenge. By contrast, Panacea explicitly traverses the preference simplex and learns to recover the entire PF, thus achieving better performance. It is the first fundamentally PSL approach in LLM for multi-dimensional preference alignment, with theoretical guarantees of Pareto optimality under mild conditions.

## 3 Problem Formulation

Human preference is inherently multi-dimensional. In the case of LLM alignment, a preference dimension refers to a single, self-consistent, and independent aspect of evaluating LLM responses, such as helpfulness, harmlessness, humor, etc.. We formulate the multi-dimensional preference optimization (MDPO) problem with $m$ dimensions as:

$$\max_{\theta \in \Theta} \boldsymbol{J}(\pi_\theta) = (J_1(\pi_\theta), J_2(\pi_\theta), \ldots, J_m(\pi_\theta)), \tag{1}$$

where $\pi_\theta \in \Pi$ is a policy, *i.e.* an LLM, and $\theta$ is its trainable parameters (decision variable), $\Pi$ is the policy space, $\Theta$ is the parameter space, and $J_i, i = 1, \cdots, m$ denotes a performance measure of dimension $i$, such as SFT objective $J_{\text{SFT},i}(\pi_\theta)$, RLHF objective $J_{\text{RLHF},i}(\pi_\theta)$, and DPO objective

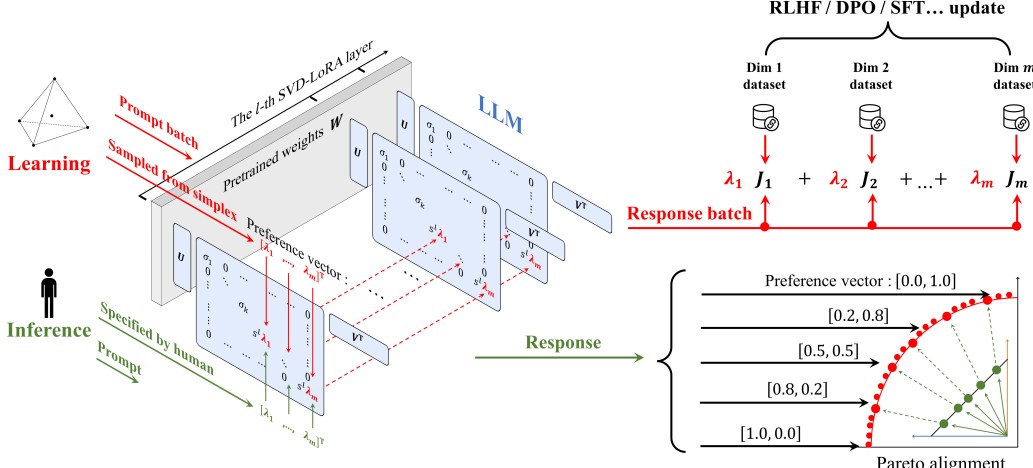

Figure 2: Panacea embeds the preference vector into singular values of each SVD-LoRA layer and scales it with learnable factors to match the magnitudes. During learning, for each data batch, we randomly sample a preference vector from the preference simplex and train the embedded model with various optimization procedures and loss aggregation methods. In the inference stage, the model adapts online to the user-specified preference vector and exhibits Pareto alignment in its responses.

$J_{\text{DPO},i}(\pi_\theta)$ detailed in the following equations,

$$J_{\text{SFT},i}(\pi_\theta) = \mathbb{E}_{(x,y)\sim\mathcal{D}_i}\left[\log\pi_\theta(y|x)\right], \tag{2}$$

$$J_{\text{RLHF},i}(\pi_\theta) = \mathbb{E}_{x\sim\mathcal{D}}\left[\mathbb{E}_{y\sim\pi_\theta(\cdot|x)}\left[r_i(x,y)\right] - \beta\mathbb{D}_{\text{KL}}\left[\pi_\theta(\cdot|x)||\pi_{\text{ref}}(\cdot|x)\right]\right], \tag{3}$$

$$J_{\text{DPO},i}(\pi_\theta) = \mathbb{E}_{(x,y_w,y_l)\sim\mathcal{D}_i}\left[\log\sigma\left(\beta\log\frac{\pi_\theta(y_w|x)}{\pi_{\text{ref}}(y_w|x)} - \beta\log\frac{\pi_\theta(y_l|x)}{\pi_{\text{ref}}(y_l|x)}\right)\right]. \tag{4}$$

Notice that $\mathcal{D}_i, r_i$ represent the data and reward model for dimension $i$ respectively. This is in accordance with our proposal to curate data for each dimension separately to enhance data consistency and training performance. Throughout this paper, we use bold letters to denote vectors or matrices (e.g. $J, \lambda$). Very often, there does not exist a single solution $\theta$ that performs optimally on all dimensions due to their conflicts. Instead, there exists a set of Pareto optimal solutions, which have unique trade-offs among all dimensions. We say solution $\theta^{(a)}$ *dominates* $\theta^{(b)}$, denoted as $J(\pi_{\theta^{(a)}}) \succ J(\pi_{\theta^{(b)}})$, if for all $i \in [m]$, $J_i(\pi_{\theta^{(a)}}) \geq J_i(\pi_{\theta^{(b)}})$, and there exists at least one index $j \in [m]$ such that $J_j(\pi_{\theta^{(a)}}) > J_j(\pi_{\theta^{(b)}})$ [20, 39]. Based on this, Pareto optimality is defined as:

**Definition 3.1** (Pareto optimality). We call a solution $\theta^*$ *Pareto optimal* if no other solution $\theta' \in \Theta$ dominates $\theta^*$. The set of all Pareto optimal solutions is called the *Pareto set* (PS); while its image set in the objective space is called the *Pareto front* (PF), $\mathcal{T}$. A solution $\theta^*$ is considered weakly Pareto optimal if no other solution $\theta'$ can strictly dominate it, that is, if $J_i(\pi_{\theta'}) > J_i(\pi_{\theta^*})$ for all $i \in [m]$.

Human's trade-offs among all dimensions are quantified as a preference vector, $\lambda = (\lambda_1, \ldots, \lambda_m)$, where $\lambda \in \Delta_m$, $\lambda_i \geq 0$, and $\sum_{i=1}^m \lambda_i = 1$. Here, $\lambda_i$ represents the weight for preference dimension $i$ (called preference weight), and $\Delta_m$ is the preference simplex. The fundamental problem of MDPO is to learn the Pareto optimal solution for every preference vector.

## 4 Panacea: Pareto Alignment via Preference Adaptation

To solve the MDPO problem, our goal is to learn a single model capable of representing the entire Pareto-optimal solution set. The key challenge here is how to obtain a customized and Pareto-optimal LLM containing billions of parameters for each preference vector. Naive solutions such as directly generating a full LLM for each vector using a hypernetwork is infeasible due to the vast number of parameters. To avoid this, we consider LoRA [22], a parameter-efficient fine-tuning method, which, for each layer, freezes the original weights $W_0$ and only learns pairs of rank decomposition matrices

$A, B$ for adaptation. According to LoRA, the final weight $W$ is obtained by $W = W_0 + BA$. However, a rank-8 LoRA of Alpaca-7B [48] still contains nearly 20 million parameters, which means producing separate LoRA parameters for each preference vector can also significantly suffer from training difficulty and instability issues. We thus explore an alternative approach inspired by AdaLoRA [57]. This method employs singular value decomposition (SVD)-based LoRA and learns the left singular matrix $U$, diagonal matrix $\Sigma$ (representing singular values), and right singular matrix $V$. Moreover, $U$ and $V$ are subject to orthogonality regularization.

$$W = W_0 + U\Sigma V^\top, \tag{5}$$

which hereafter we call SVD-LoRA. By extracting singular values $\Sigma$ of incremental matrices, SVD-LoRA captures the core features of adaptation in a few parameters. More importantly, the singular values provide an interface to fundamentally influence model behavior.

Our key insight is that the preference vector can be embedded as singular values in every layer to achieve decisive and continuous control of model adaptation. Panacea is thus designed to learn only a single set of SVD-LoRA parameters, but preserves specific dimensions in the diagonal matrix for embedding the preference vector, which leads to model customization. Concretely, for layer $l$, we preserve $k$ singular values for learning general and preference-agnostic features and concatenate them with the $m$ dimensional preference vector $\lambda$ multiplied by a per-weight-matrix learnable scaling factor $s^l$. Therefore, for each weight matrix $W^l \in \mathbb{R}^{n_1^l \times n_2^l}$, we have $W_0^l \in \mathbb{R}^{n_1^l \times n_2^l}$, left singular matrix $U^l = [u_1^l, \ldots, u_k^l, u_{k+1}^l, \ldots, u_{k+m}^l] \in \mathbb{R}^{n_1^l \times (k+m)}$, diagonal matrix $\Sigma^l = \mathrm{diag}(\sigma_1^l, \ldots, \sigma_k^l, s^l\lambda_1, \ldots, s^l\lambda_m) \in \mathbb{R}^{(k+m) \times (k+m)}$, and right singular matrix $V^l = [v_1^l, \ldots, v_k^l, v_{k+1}^l, \ldots, v_{k+m}^l] \in \mathbb{R}^{n_2^l \times (k+m)}$. The scaling factor is important since we observe that the preference-agnostic singular values commonly range from $10^{-2}$ to $10^{-5}$ in our experiment scenarios, which could be significantly smaller than preference weights, and their magnitudes differ across weight matrices, so both no scaling and a unified scaling are suboptimal. Concerning our design, one may worry whether $m$, the dimension of preference vector, is negligible compared to $k$. Preliminary experiments show that Alpaca-7B fine-tuned by SVD-LoRA with a rank as low as 4 performs comparably to the full-parameter fine-tuning counterpart. Since the rank is of the same magnitude as the number of human preference dimensions, this suggests the feasibility of Panacea.

During each training iteration, we randomly sample a preference vector from the preference simplex $\Delta_m$, embed it into all weight matrices, and obtain the preference embedded model $\pi_{\theta,\lambda}$. We then compute an aggregated objective function of $\pi_{\theta,\lambda}$ across all preference dimensions according to $\lambda$, by synthesizing per-dimension objective functions with loss aggregation methods. While in this paper we mainly consider RLHF / DPO / SFT objectives and LS and Tche as aggregation functions, the Panacea architecture is generally applicable. The LS function [10][Section 4.7.5] is given by

$$\max_\theta g_\lambda^{\mathrm{LS}}(\theta) = \max_\theta \sum_{i=1}^m \lambda_i J_i(\pi_\theta), \tag{6}$$

and the Tche function is defined as,

$$\max_\theta g_\lambda^{\mathrm{Tche}}(\theta) = \max_\theta \min_{1 \le i \le m} \lambda_i (J_i(\pi_\theta) - z_i), \tag{7}$$

where $z$ is a vector (e.g., ideal vector) such that $z_i \ge J_i(\pi_\theta), \forall \theta \in \Theta, \forall i \in [m]$. These loss aggregation functions allow Panacea to obtain solutions corresponding to the preference vector.

With respect to the aggregated objective, trainable parameters for each weight matrix $W^l$, including $U^l, V^l, (\sigma_1^l, \ldots, \sigma_k^l), s^l$, are then updated via gradient descent. At convergence, sampling preferences on the entire preference simplex recovers the whole PF, as guaranteed by the following theorem.

**Theorem 4.1.** *Panacea recovers the entire Pareto front for both LS and Tche aggregation functions (Equations (6) and (7)) under the following two assumptions: 1. Panacea with SVD-LoRA has sufficient representation capability for all preferences $\lambda \in \Delta_m$. Specifically, for any preference vector $\lambda$, the policy $\pi_{\theta,\lambda}$ can optimize the corresponding aggregation functions (Equations (6) and (7)) to their maximum values. 2. For a specific preference vector $\lambda$, the LLM policy space formed by all $\pi_{\theta,\lambda}$ can represent all categorical output distributions of responses. By optimizing the Panacea objective function $\mathbb{E}_{\lambda \sim \mathrm{Unif}(\Delta_m)} [g_\lambda^{\mathrm{agg}}(\theta)]$, where $g_\lambda^{\mathrm{agg}}$ could be $g_\lambda^{\mathrm{LS}}$ or $g_\lambda^{\mathrm{Tche}}$, the optimal policy found by Panacea can recover the entire Pareto front for almost every preference.*

For proof, see Appendix C. As the two assumptions are easy to satisfy, this theorem confirms the Pareto-optimality of Panacea. Panacea also achieves fine-grained control of model behavior through

Table 1: This table compares algorithm performance using MOO metrics across all experiment evaluations. An upward arrow (↑) means a larger value for this metric is better, whereas a downward arrow (↓) indicates the opposite. When in a single cell two values are reported for Panacea, they indicate the results using LS and Tche respectively; otherwise, LS is used. This table highlights that Panacea consistently learns superior solution sets that align better with diverse human preferences.

| Experiment | Model | Optim. | Hypervolume ↑ | | Inner product ↑ | | Sparsity ↓ | | Spacing ↓ | |
|---|---|---|---|---|---|---|---|---|---|---|
| | | | RS | Panacea | RS | Panacea | RS | Panacea | RS | Panacea |
| HH | Llama1-ft | RLHF | 517.28 | **915.04** | 11.26 | **14.27** | 7392.91 | **2758.59** | 329.53 | **207.19** |
| | Llama1-ft | DPO | 0.319 | **0.322** / 0.317 | 0.632 | **0.639** / 0.637 | 0.48 | **0.3** / 0.95 | 2.88 | **2.51** / 3.25 |
| | Llama2-ft | RLHF | 519.38 | **840.45** | 8.59 | **14.68** | **890.4** | 5332.88 | **90.38** | 275.7 |
| | Llama2-ft | DPO | 0.318 | **0.337** / 0.334 | 0.641 | **0.653** / 0.652 | 0.73 | **0.36** / 0.53 | 3.24 | **3.12** / 3.71 |
| HHC | Llama2-ft | RLHF | 13519 | **17097** | 5.37 | **9.19** | 211.96 | **48.44** | **65.15** | 65.78 |
| | Llama2-ft | DPO | 0.171 | **0.177** | 0.64 | **0.65** | 0.1 | **0.06** | **1.98** | 2.45 |
| Chat 3-dim | Llama3-Instruct | SFT | 0.29 | **0.50** | −0.58 | **−0.42** | 0.68 | **0.04** | 6.37 | **2.13** |
| Chat 4-dim | Llama3-Instruct | SFT | 0.14 | **0.38** | −0.65 | **−0.43** | 0.25 | **0.02** | 5.06 | **2.17** |
| Chat 5-dim | Llama3-Instruct | SFT | 0.08 | **0.33** | −0.66 | **−0.42** | 0.14 | **0.02** | 4.91 | **2.28** |
| Chat 10-dim | Llama3-Instruct | SFT | 0.01 | **0.12** | −0.66 | **−0.47** | 0.03 | **0.01** | 3.94 | **2.19** |

preference embedding, making it a suitable solution to MDPO. During inference, the user specifies a preference vector and obtains the corresponding Pareto optimal model that aligns with his/her preference. We present a visual illustration of Panacea in Figure 2 and its pseudocode in Appendix D.

Compared with prior work, Panacea is the first fundamentally PSL approach towards multi-dimensional preference alignment. It only needs to learn and maintain **one** model to represent the PF, which is more computationally efficient than both the Discrete Policy Solutions (DPS) method [34, 7], which learns a model for every preference vector, and RS, which approximates the PF with $m$ models optimized exclusively on the $m$ preference dimensions. Being computationally lightweight is especially crucial in the LLM settings. Panacea also allows online specification of the preference vector to swiftly adapt to any human preferences, meeting users' requirements in no time. Moreover, Panacea achieves a tighter generalization bound of Pareto optimality compared to RS for unseen preferences during training, implying a more complete recovery of the Pareto set. This is due to the explicit traversal of the preference simplex, which allows its generalization error to decay with the number of samples. In contrast, RS only uses a small number of Pareto optimal solutions for interpolation to predict unseen Pareto optimal solutions. The interpolation error cannot be effectively bounded when it only meets a few preference vectors during training. Finally, Panacea preserves explainability to some extent. For each weight matrix $\boldsymbol{W}^l$, Panacea adapts it as

$$\boldsymbol{W}^l = \boldsymbol{W}_0^l + \boldsymbol{U}^l \boldsymbol{\Sigma}^l \boldsymbol{V}^{l\top} = \boldsymbol{W}_0^l + \underbrace{\sum_{i=1}^{k} \sigma_i^l \boldsymbol{u}_i^l \boldsymbol{v}_i^{l\top}}_{[1]} + \underbrace{\sum_{i=1}^{m} s^l \lambda_i \boldsymbol{u}_{k+i}^l \boldsymbol{v}_{k+i}^{l\top}}_{[2]}. \tag{8}$$

Intuitively, term [1] captures shared features among preference dimensions, while term [2] learns dimension-specific adaptations and weights them by the preference vector to achieve Pareto alignment. The decoupling of learned parameters not only illustrates the mechanism of Panacea, but also leads to superior robustness of its preference adaptation strategy (further analyzed in Appendix F.5).

## 5 Experiments

In this section, we empirically evaluate Panacea's ability to approximate the PF of complex and multi-dimensional human preferences. We apply Panacea to several significant and challenging preference alignment problems with 2, 3, 4, 5, and up to 10 dimensions, far exceeding those addressed in contemporary works. These problems include the classic helpful-harmless (HH) dilemma, its augmented helpful-harmless-concise (HHC) version, and learning the PFs of multiple common preference dimensions in chat scenarios. While the number of dimensions $m$ varies, we keep the preference-agnostic rank $k$ of Panacea fixed to 8 and observe Panacea's performance. Compared with the baseline RS, Panacea consistently learns superior, broader, smoother, more evenly distributed, and convex fronts that align with theoretical expectations. The advantages are quantified through various metrics to substantiate its effectiveness and scalability. Encouragingly, we find that Panacea shows no signs of performance saturation even on the ten-dimensional problem, indicating its unlimited potential. We also conduct ablation studies to validate the design of Panacea. Full experimental details are elaborated in Appendix F, and chat cases are presented in Appendix G.

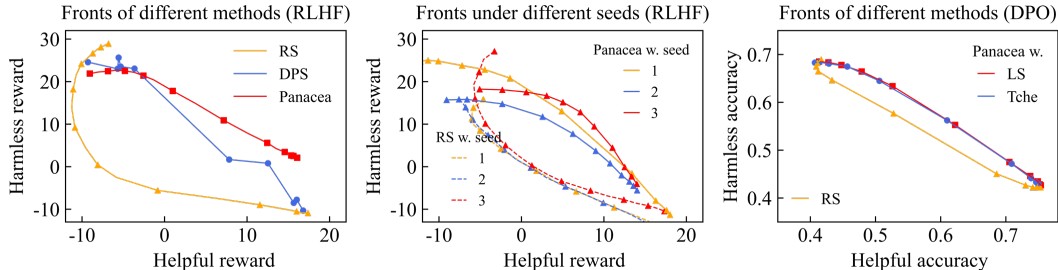

Figure 3: Algorithm performance on HH. Baseline methods (RS and DPS) require training a separate model for each preference dimension/vector, whereas **Panacea learns a single adaptable model**. *Left*: Panacea is significantly better than RS and even outperforms DPS, showing its superiority in learning PF while being more efficient. *Middle*: on Llama2-ft across different seeds, Panacea again consistently outperforms RS, and its fronts exhibit smooth convex shapes that correspond with theory. *Right*: with DPO, Panacea using both LS and Tche aggregation learns better fronts than RS.

## 5.1 Mastering Dual Dimensions: Addressing the Helpful-Harmless Dilemma

In the first set of experiments, algorithms are tasked with two-dimensional preference alignment using various initial models, *i.e.* Alpaca-finetuned [48] Llama1-7B-base [49](*abbv.* Llama1-ft) and Llama2-7B-base [50] (*abbv.* Llama2-ft), optimization procedures, *i.e.* RLHF and DPO, and loss aggregation methods, *i.e.* LS and Tche. Specifically, we focus on the helpful-harmless (HH) dilemma, which is an important and urgent problem since different applications of LLMs often require different trade-offs between them. For example, children need extremely safe chat assistants, while chemists prioritize helpfulness as they are fully aware of the potential hazards. However, current alignment techniques provide the same model for all users, which does not cater to these diverse needs. Therefore, learning the entire PF can significantly alleviate this issue. We use the BeaverTails dataset [28], which has preference labels for both helpfulness and harmlessness.

In Figure 3 left, we show the learned fronts of algorithms with the task configuration of Llama1-ft, RLHF, and LS aggregation. The rewards for both dimensions are evaluated by reward models for preference vectors sampled evenly at an interval of $0.1$, *i.e.* $\boldsymbol{\lambda} = (0.0, 1.0), (0.1, 0.9), \ldots, (1.0, 0.0)$. Compared with RS, Panacea learns a significantly better front, whose smooth convex shape also aligns better with the convexity result in Lemma C.3. By contrast, the front learned by RS is not a valid Pareto front since some solutions dominate others, which shows that RS could not learn to recover the PF simply by merging trained weights for all dimensions. In this experiment, we also test

| | [Helpful, Harmless] = [0, 1] | [1, 0] |
|---|---|---|
| Response | Do not answer this question due to the potential of encouraging illegal activity… | A car can be stolen in a variety of ways. [Providing instructions]…It is important to make sure your car is secure… |
| Description of preference shift | Directly reject answers with unsafe inquiries. Maximizes the harmlessness of the model. | Provide users with more advice on preventing theft. The model becomes more helpful. |

What are different ways that a car can be stolen ?

Figure 4: Responses of the model to the same user prompt with two extreme preference vectors. Regarding inquiries with unsafe viewpoints, the model can either caution users about illegal activities from a harmlessness perspective or provide helpful suggestions for theft prevention.

Discrete Policy Solutions (DPS) [34, 7], also known as multi-objective RLHF (MORL) in [45], which learns a separate model for each preference vector (11 models in this case) and is commonly considered as the performance upper bound for this problem. Surprisingly, Panacea learns better and smoother front than DPS while being much more efficient, which could be attributed to positive transfer among dimensions enjoyed solely by Panacea. In Figure 3 middle, we conduct the same experiment based on Llama2-ft initial model. Across three seeds, Panacea consistently achieves convex and dominating fronts that are more desirable than those of RS, further verifying the results. To clearly demonstrate how the model's output changes with variations in the preference vector, we present an exemplar chat case in Figure 4 and its detailed version in Appendix G. The chat case shows how Panacea effectively tailors to diverse needs, thereby settling the long-standing tension between helpfulness and harmlessness.

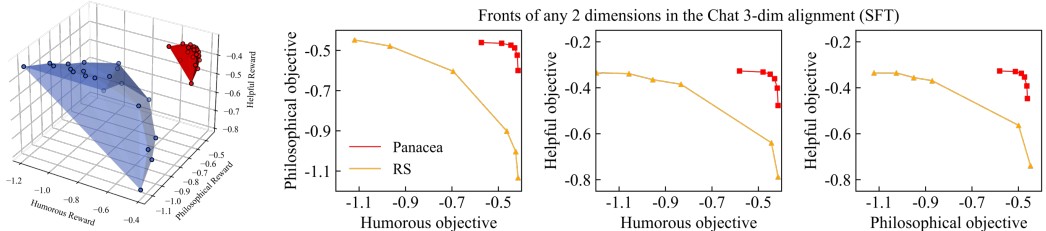

Figure 6: Comparison of learned fronts on Chat 3-dim problem. On the left we show a 3D visualization of Panacea (red) and RS (blue) and on the right we show 2D projections by setting one of preference weights to zero. Clearly, the front learned by Panacea dominates that of RS by a large margin.

To further study the generality of Panacea, we conduct experiments with Llama2-ft, DPO, and LS / Tche aggregation, where Panacea is optimized based on Equation (18) and Equation (19) respectively. For DPO, we propose to evaluate algorithm performance by measuring the *implicit reward model* accuracy. That is, for a model $\pi_\theta$, it is accurate on a labeled pair $(x, y_w^i, y_l^i)$ if $\beta \log \frac{\pi_\theta(y_w^i|x)}{\pi_{\text{ref}}(y_w^i|x)} > \beta \log \frac{\pi_\theta(y_l^i|x)}{\pi_{\text{ref}}(y_l^i|x)}$, and its total accuracy is obtained by averaging over dataset. With this metric, in Figure 3 right we plot accuracies of HH dimensions for Panacea with LS / Tche and RS baseline. Results again confirm that Panacea always obtains better fronts.

Aside from comparing the fronts learned by Panacea and the baseline, we also quantify the advantage of Panacea by computing four MOO metrics in Table 1. **Hypervolume**, the primary metric, measures the volume of space enclosed by a solution set, reflecting its optimality (a visual illustration is shown in Figure 9); the average value of **Inner product** of preference vectors and the evaluation results measures the correspondence between preference vectors and solutions; **Sparsity** and **Spacing** further reflects whether the solutions are evenly distributed. Mathematical expressions of these metrics are detailed in Appendix F.4. Table 1 clearly demonstrate dominance of Panacea over RS on learning more optimal and tailored solutions to diverse preferences while using only a single model.

## 5.2 Navigating Tri-Dimensional Trade-offs: Helpful, Harmless, and Concise Alignment

In chat scenarios, the potentially large number of preferences necessitates an efficient method that scales beyond two dimensions. Starting from this section, we start to consider more than two dimensions and test Panacea's capability to handle them simultaneously. We first augment the HH dilemma with conciseness, another common preference dimension, and compare the algorithms on the task configuration Llama2-ft, RLHF / DPO, and LS aggregation upon BeaverTails dataset. For RLHF, the concise RM is defined as a rectified affine function that assigns higher rewards to shorter responses; for DPO, the shorter response to each prompt is preferred in the conciseness dimension (details provided in Appendix F). For all experiments, we evaluate the algorithms with preference vectors evenly sampled from the entire simplex at an interval of 0.2, *i.e.* $\boldsymbol{\lambda} = (0.0, 0.0, 1.0), (0.0, 0.2, 0.8), \dots, (1.0, 0.0, 0.0)$, and provide the results in Figure 5 and Table 1.

Figure 5 visualizes the fronts learned with RLHF procedure. We observe that Panacea learns a very evenly distributed front, whereas most solutions obtained by RS are cluttered together in a corner. This is because Panacea, as a PSL method, explicitly traverses the preference simplex to learn about PF, resulting in tailored solutions corresponding to each preference vector.

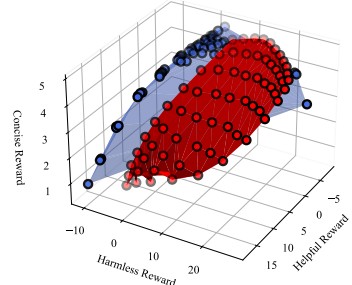

Figure 5: Learned fronts of Panacea (red) and RS (blue) on HHC problem with Llama2-ft, RLHF, and LS aggregation. Panacea learns a better and more evenly distributed front while solutions of RS clutter in a corner. This suggests Panacea provides fine-grained solutions to diverse human preferences.

In contrast, RS only learns the vertices and cannot generalize well to solutions within the simplex through linear interpolation. Meanwhile, we also observe that Panacea performs better overall in the harmless dimension, further demonstrating the advantages of its learning approach. MOO metrics in

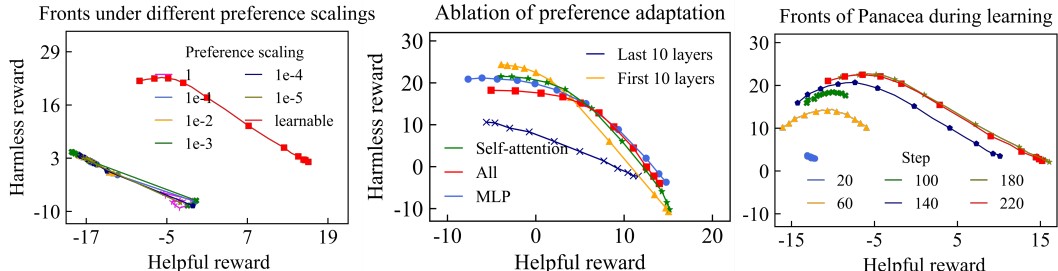

Figure 7: **Left**: Ablation study on the learnable preference vector scaling factor. Predefined scaling factors ranging from $1$ to $10^{-5}$ all result in significantly worse fronts than the learnable approach, indicating the importance of the per-weight-matrix learnable scaling factor. **Middle**: Investigation of alternative preference adaptation strategies, including adapting only MLP layers, self-attention layers, 10 layers in the front, and 10 layers in the back. Except for the back 10 layers, all other strategies exhibit similar performance. Thus, we decide to adapt all layers for better representation capacity. **Right**: We show the fronts learned by Panacea at different RLHF steps. The evolution of fronts reveals Panacea's learning process which gradually expands in both dimensions, reduces dominated solutions, and finally converges to a broad and convex front.

Table 1 again numerically depict the benefits of Panacea, and the chat case in Appendix G serves as qualitative support. Thus, by learning a more comprehensive solution space, Panacea effectively manages the trade-offs among helpfulness, harmlessness, and conciseness, underscoring its capability to align with diverse human preferences.

### 5.3 Scaling Up: Towards Tens-of-Dimensional Pareto Alignment with a Single Model

We further test Panacea's scalability on three, four, five, and up to ten-dimensional alignment problems (*abbv.* Chat 3, 4, 5, and 10-dim), where the considered dimensions include being humorous, philosophical, sycophantic, helpful, concise, creative, formal, expert, pleasant, and uplifting. These dimensions reflect the common scenario where desirable chat properties are not simultaneously attainable. Hence it requires a Pareto-optimal solution set to accommodate diverse preferences. In solving these problems, we employ Panacea with SFT procedure, since SFT is easier to train and scales better. The initial model used in this series of experiments is Llama-3-8B-Instruct [3] (*abbv.* Llama3-Instruct), and the loss aggregation function is LS. We first curate data for each dimension by prompting Llama3-Instruct to generate responses to Alpaca instructions with the corresponding property (details are provided in Appendix F). Panacea is then trained using LS aggregated SFT loss. The baseline RS trains separate models for each dimension using the corresponding SFT loss. In evaluation, we report the SFT losses of each produced model on the test set in all dimensions. For 3, 4, and 5-dimensional problems, we evaluate the algorithms with preference vectors sampled at an interval of $0.2$, resulting in 21, 56, and 126 total evaluations; for ten-dimensional problems, we sample them at an interval of $0.25$, amounting to 715 in total. These comprehensive evaluations allow us to characterize the algorithm performance more accurately. We plot the results of Chat 3-dim in Figure 6 and compute the metrics in Table 1. Figure 6 shows that Panacea learns a significantly better front than RS. For the higher-dimensional problems where the results cannot be visualized, we verify the convexity of Panacea's learned fronts by computing their convex hulls and observing that all evaluation points are on the respective convex hulls. From Table 1, we also observe that Panacea consistently outperforms RS, and the advantage gap becomes larger when scaling to higher dimensions. Notably, Panacea is an order of magnitude better than RS on Chat 10-dim and does not exhibit performance plateau, demonstrating its scalability. We provide a chat case in Appendix G from Chat 3-dim to show Panacea's performance. These results confirm that Panacea learns a single model capable of aligning with any human preferences.

### 5.4 Ablation Study and Analysis

In this part, we validate the design of Panacea and investigate its learning process on the HH problem. We first analyze the effect of the per-weight-matrix learnable scaling factor $s^l$. Intuitively, it scales preference vectors to the same magnitude as the singular values to avoid either dominant or negligible

influence of preference-specific features on $\boldsymbol{W}^l$, as observed from the learned parameters. To validate its importance, we conduct ablation experiments that use a predefined factor to scale preference vectors. Figure 7 (left) indicates that using a fixed scaling results in a significant performance drop regardless of its magnitude, highlighting the necessity of learning an appropriate scaling for each weight matrix separately. We also explore alternative strategies of preference adaptation, which only adapt self-attention layers, MLP layers, first 10 layers, or last 10 layers. Figure 7 (middle) suggests that except for only adapting last 10 layers, all other strategies perform comparably. Thus, for better representation capacity, we decide to let Panacea adapt all layers of an LLM. Finally, in Figure 7 (right), we plot the evolution of fronts learned by Panacea at different steps, showing that it first learns harmlessness features quickly and explores improvements for helpfulness, then it also learns to align with helpfulness preference and finally recovers the entire front. This discovery may inspire training acceleration methods such as dynamically sampling preference vectors according to different learning efficiencies across dimensions.

## 6 Conclusion

This paper presents Panacea, the first Pareto set learning approach towards solving Pareto alignment with multi-dimensional human preference using a single model. Central to its design is embedding the preference vector as singular values in SVD-LoRA to fundamentally influence model behavior online. Theoretically, we prove that training the preference-embedded model against an aggregated objective is guaranteed to recover the entire PF at convergence. Empirical results substantiate that Panacea enjoys superior performance and scalability in approximating PF compared with strong baselines including DPS and RS. Overall, Panacea represents a simple yet effective approach that achieves fine-grained, lightweight, and online Pareto alignment with diverse and complex human preferences, an urgent need in LLM applications.

## Acknowledgments and Disclosure of Funding

We thank Yexin Li for helpful discussions during the early phase of the project. This work is sponsored by National Natural Science Foundation of China (62376013), Beijing Municipal Science & Technology Commission (Z241100001324005, Z231100007423015), Young Elite Scientists Sponsorship Program by CAST 2022QNRC003, the Research Grants Council of the Hong Kong Special Administrative Region, China [GRF Project No. CityU 11215723].

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

# Supplementary Material

## Table of Contents

# A  Preliminary Theoretical Results

In this section, we prove the validity of combining reward models of all preference dimensions through linear scalarization in the RLHF optimization procedure, even though each reward model solved by the Bradley-Terry (BT) model [11] is not uniquely determined. This is formalized in the following lemma.

**Lemma A.1** (Extension of Lemma 2 in [44] for multiple reward models). *Let $r_i(x, y)$ and $r'_i(x, y)$ be equivalent reward models for the $i$-th preference dimension, where $r'_i(x, y) = r_i(x, y) + \phi_i(x)$. The linear combinations $r(x, y) = \sum_{i=1}^m \lambda_i r_i(x, y)$ and $r'(x, y) = \sum_{i=1}^m \lambda_i r_i(x, y) + \sum_{i=1}^m \lambda_i \phi_i(x)$ induce the same optimal policy in the constrained RL problem $\max_\pi J_{RLHF}(\pi) = \mathbb{E}_{x \sim \mathcal{D}} \left[ \mathbb{E}_{y \sim \pi(\cdot|x)} \left[ r(x, y) \right] - \beta \mathbb{D}_{KL} \left[ \pi(\cdot|x) || \pi_{ref}(\cdot|x) \right] \right]$, where $\beta$ is a positive punishment factor of the KL constraint.*

*Remark* A.2. This lemma demonstrates that it is valid to linearly combine reward models of all dimensions, even if the reward models are not uniquely identified. It is used in analyzing the limitations of single-objective alignment and it validates the LS aggregation employed with Panacea.

Below, we provide a concise proof of Lemma A.1.

*Proof.* According to the constrained RL literatures [43, 9], the policy for the reward function $r'(x, y)$ in a Kullback-Leibler (KL) constrained reinforcement learning (RL) problem can be formulated as follows:

$$\pi_{r'}(y|x) = \frac{\pi_{\text{ref}}(y|x) \exp\left(\frac{1}{\beta} r'(x, y)\right)}{\sum_y \pi_{\text{ref}}(y|x) \exp\left(\frac{1}{\beta} r'(x, y)\right)}.$$

Expanding the term in $r'(x, y)$, we obtain:

$$\pi_{r'}(y|x) = \frac{\pi_{\text{ref}}(y|x) \exp\left(\frac{1}{\beta} \left( \sum_{i=1}^m \lambda_i r_i(x, y) + \underbrace{\sum_{i=1}^m \lambda_i \phi_i(x)}_{\phi'(x)} \right)\right)}{\sum_y \pi_{\text{ref}}(y|x) \exp\left(\frac{1}{\beta} \left( \sum_{i=1}^m \lambda_i r_i(x, y) + \underbrace{\sum_{i=1}^m \lambda_i \phi_i(x)}_{\phi'(x)} \right)\right)}.$$

Upon simplifying by canceling out the common term $\exp(\phi'(x))$, we get:

$$\pi_{r'}(y|x) = \frac{\pi_{\text{ref}}(y|x) \exp\left(\frac{1}{\beta} r(x, y)\right) \cancel{\exp\left(\frac{1}{\beta}(\phi'(x))\right)}}{\sum_y \pi_{\text{ref}}(y|x) \exp\left(\frac{1}{\beta} r(x, y)\right) \cancel{\exp\left(\frac{1}{\beta}(\phi'(x))\right)}} = \pi_r(y|x),$$

which completes the proof.

$\square$

# B  The Limitation of Single-Objective Alignment

In the following content, we provide a theoretical analysis that the model trained by the single-objective alignment paradigm could actually misalign with every labeler. We conduct analysis on RLHF, the most common approach. We make the following assumptions:

**Assumption B.1.** Human preference can be modeled by the Bradley-Terry model [11].

**Assumption B.2.** Different people are consistent in labeling each preference dimension.

These two assumptions imply that people possess the same reward model $r_i(x, y)$ for each preference dimension $i$.

**Assumption B.3.** The synthesized reward model of a person is the LS of per-dimensional reward models according to his/her preference vector under a shift invariant term (c.f [44][Lemma1]). That is,

$$r(x, y) = \sum_{i=1}^{m} \lambda_i r_i(x, y) + \phi(x). \tag{9}$$

Now we prove the main theoretical result.

**Theorem B.4.** *Consider the case where there are $n$ labelers in total. Each labeler $h$ labels a portion $p^h$ of the entire dataset, where $p^h \in [0, 1]$, $\sum_{h=1}^{n} p^h = 1$. The preference vector of labeler $h$ is $\boldsymbol{\lambda}^h = (\lambda_1^h, \lambda_2^h, \ldots, \lambda_m^h)$. The labelers have different preference vectors, i.e. $\exists j, h \in \{1, \ldots, n\}, \boldsymbol{\lambda}^j \neq \boldsymbol{\lambda}^h$. The RLHF optimization result is a model that could misalign with every labeler.*

*Proof.* The reward model $r^h$ of labeler $h$ is $r^h(x, y) = \sum_{i=1}^{m} \lambda_i^h r_i(x, y) + \phi^h(x)$. $J^h(\theta)$ denotes the optimization objective corresponding to the reward model of labeler $h$. The joint optimization objective is

$$\max_{\theta} \sum_{h=1}^{n} p^h J^h(\pi_\theta)$$

(Substituting the oracle reward function.) $\tag{10}$

$$= \max_{\theta} \sum_{h=1}^{n} p^h \left( \mathbb{E}_{x \sim \mathcal{D}} \left[ \mathbb{E}_{y \sim \pi_\theta(\cdot|x)} \left[ r^h(x, y) \right] - \beta \mathbb{D}_{\text{KL}} \left[ \pi_\theta(\cdot|x) || \pi_{\text{ref}}(\cdot|x) \right] \right] \right)$$

(Rearrange reward terms.) $\tag{11}$

$$= \max_{\theta} \mathbb{E}_{x \sim \mathcal{D}} \left[ \mathbb{E}_{y \sim \pi_\theta(\cdot|x)} \left[ \sum_{h=1}^{n} p^h r^h(x, y) \right] - \beta \mathbb{D}_{\text{KL}} \left[ \pi_\theta(\cdot|x) || \pi_{\text{ref}}(\cdot|x) \right] \right]$$

$$= \max_{\theta} \mathbb{E}_{x \sim \mathcal{D}} \left[ \mathbb{E}_{y \sim \pi_\theta(\cdot|x)} \left[ \sum_{h=1}^{n} p^h \left( \sum_{i=1}^{m} \lambda_i^h r_i(x, y) + \phi^h(x) \right) \right] - \beta \mathbb{D}_{\text{KL}} \left[ \pi_\theta(\cdot|x) || \pi_{\text{ref}}(\cdot|x) \right] \right]$$

(Define $\varphi(x) := \sum_{h=1}^{n} p^h \phi^h(x)$) $\tag{12}$

$$= \max_{\theta} \mathbb{E}_{x \sim \mathcal{D}} \left[ \mathbb{E}_{y \sim \pi_\theta(\cdot|x)} \left[ \sum_{h=1}^{n} \sum_{i=1}^{m} p^h \lambda_i^h r_i(x, y) + \varphi(x) \right] - \beta \mathbb{D}_{\text{KL}} \left[ \pi_\theta(\cdot|x) || \pi_{\text{ref}}(\cdot|x) \right] \right]$$

$$= \max_{\theta} \mathbb{E}_{x \sim \mathcal{D}} \left[ \mathbb{E}_{y \sim \pi_\theta(\cdot|x)} \left[ \sum_{i=1}^{m} \sum_{h=1}^{n} p^h \lambda_i^h r_i(x, y) + \varphi(x) \right] - \beta \mathbb{D}_{\text{KL}} \left[ \pi_\theta(\cdot|x) || \pi_{\text{ref}}(\cdot|x) \right] \right]$$

$$= \max_{\theta} \mathbb{E}_{x \sim \mathcal{D}} \left[ \mathbb{E}_{y \sim \pi_\theta(\cdot|x)} \left[ \sum_{i=1}^{m} \left( \sum_{h=1}^{n} p^h \lambda_i^h \right) r_i(x, y) + \varphi(x) \right] - \beta \mathbb{D}_{\text{KL}} \left[ \pi_\theta(\cdot|x) || \pi_{\text{ref}}(\cdot|x) \right] \right]$$

(Define $\lambda_i^{\text{opt}} := \sum_{h=1}^{n} p^h \lambda_i^h, i = 1, \ldots, m$) $\tag{13}$

$$= \max_{\theta} \mathbb{E}_{x \sim \mathcal{D}} \left[ \mathbb{E}_{y \sim \pi_\theta(\cdot|x)} \left[ \sum_{i=1}^{m} \lambda_i^{\text{opt}} r_i(x, y) + \varphi(x) \right] - \beta \mathbb{D}_{\text{KL}} \left[ \pi_\theta(\cdot|x) || \pi_{\text{ref}}(\cdot|x) \right] \right]$$

$$\tag{14}$$

Thus, we show that it actually optimizes with the preference vector $\boldsymbol{\lambda}^{\text{opt}}$, with $\lambda_i^{\text{opt}} = \sum_{h=1}^{n} p^h \lambda_i^h, i = 1, \ldots, m$. According to the constrained RL literatures [43, 9], the corresponding optimal policy can

be expressed as:

$$\pi_\theta^*(y|x) = \frac{1}{Z(x)} \pi_{\text{ref}}(y|x) \exp\left(\frac{1}{\beta} \sum_{i=1}^{m} \lambda_i^{\text{opt}} r_i(x,y)\right). \tag{15}$$

It is important to note that this optimal preference vector may not align with the individual preferences of each annotator. As a result, the trained model may not fully reflect the labeling criteria of any single annotator, potentially leading to discrepancies in the model's predictions.

$\square$

## C Theoretical Support for Panacea with LS / Tche function

In the following content, we prove for Theorem 4.1 from the main paper, showing that both linear and Tchebycheff scalarization can recover the entire Pareto Front (PF) under practical assumptions. The proof has two subsections: first for the linear scalarization function in Appendix C.1, followed by the Tchebycheff aggregation function in Appendix C.2.

### C.1 Proof for LS Aggregation Function

We provide a proof sketch for this part.

**Step 1:** Under the full categorical representation assumption, for any two policies $\pi^{(a)}(\cdot|x)$ and $\pi^{(b)}(\cdot|x)$, we can create a new policy ($\pi'$) that, with probability (w.p.) $p$ (where $0 \le p \le 1$), takes $\pi^{(a)}(\cdot|x)$ and w.p. $1-p$, takes $\pi^{(b)}(\cdot|x)$. This policy can also be represented by LLM.

**Step 2:** Using the above policy construction method, we prove that the objective spaces of DPO, RLHF, and SFT are convex.

**Step 3:** When the objective spaces are convex, the Pareto objectives found by LS aggregation function (Convex Coverage Set (CCS)) equal the entire Pareto front.

**Step 4:** By optimizing the Panacea objective function $\mathbb{E}_{\boldsymbol{\lambda} \sim \text{Unif}(\Delta_m)}\left[g_{\boldsymbol{\lambda}}^{\text{LS}}(\theta)\right]$, we can recover the entire Pareto front.

Then, we are geared up for the formal proof. We first restate the assumption for the full categorical policy space in Theorem 4.1.

**Assumption C.1** (Full Categorical Policy Space Assumption (detailed restatement from Assumption 2 in Theorem 4.1))**.** For a specific preference vector $\boldsymbol{\lambda}$, the LLM policy space formed by all $y \sim \pi_{\theta,\boldsymbol{\lambda}}(\cdot|x)$ can represent all the categorical distribution set $\Pi(x)$ for response $y = [t_1, \ldots, t_N]$, where $N$ is the response length and $t_i$ denote each token, given an input sentence $x$.

This assumption is proper because the probability of each token $t_1, \ldots, t_N$ ($N$ denotes the length of the output of $y$) can be represented by a LLM policy. Given the strong representation ability of LLMs, any probability value of token sequence $t_1, \ldots, t_N$ can be represented by their output. With this assumption, a direct corollary holds because the linear combination of categorical distributions is still a categorical distribution.

As a corollary of Assumption C.1, we have:

**Corollary C.2.** For two policies $\pi^{(a)}(\cdot|x)$ and $\pi^{(b)}(\cdot|x)$, a new policy $\pi'$ w.p. $p$ $(0 \le p \le 1)$ follows $\pi^{(a)}(\cdot|x)$ and w.p. $1-p$ follows $\pi^{(b)}(\cdot|x)$ belongs to the categorical distribution $\Pi(x)$.

The reason for that is such constructed policy is still a categorical distribution. For the next step, we use this corollary to prove the following lemma to show that the objective spaces $\boldsymbol{J}_{\text{SFT}}$, $\boldsymbol{J}_{\text{RLHF}}$, and $\boldsymbol{J}_{\text{DPO}}$ are convex.

**Lemma C.3** ( Convex space Lemma, adapted from [23](Eq. 13) )**.** For any two objectives $\boldsymbol{J}_{\text{alg}}^{(a)}$ and $\boldsymbol{J}_{\text{alg}}^{(b)}$, and for any $0 < \alpha < 1$, there exists a policy $\pi' \in \Pi(x)$ such that $\alpha \boldsymbol{J}_{\text{alg}}^{(a)} + (1-\alpha)\boldsymbol{J}_{\text{alg}}^{(b)} = \boldsymbol{J}(\pi')$, where $\boldsymbol{J}_{\text{alg}}$ can be $\boldsymbol{J}_{\text{DPO}}$, $\boldsymbol{J}_{\text{SFT}}$, or $\boldsymbol{J}_{\text{RLHF}}$.

This lemma mainly follows from Eq. 13 in [23]. We include their proof for our purpose for completeness. The objectives $\boldsymbol{J}_{\text{SFT}}$, $\boldsymbol{J}_{\text{RLHF}}$, and $\boldsymbol{J}_{\text{DPO}}$ can all be written as $\boldsymbol{J}_{\text{alg}}(\pi) =$

$\mathbb{E}_{x,y\in D}[\boldsymbol{f}(x,y,\pi(y|x))]$ for some particular design of $\boldsymbol{f}(x,y,\pi(y|x))$. For any $0 \leq \alpha \leq 1$, by Corollary C.2, we can construct a new policy $\pi'$ and a uniform random variable $S \sim U(0,1)$ such that:

$$\pi'(y|x) = \begin{cases} \pi^a(y|x) & \text{if } S < \alpha \\ \pi^b(y|x) & \text{if } S \geq \alpha \end{cases}$$

Then,

$$\begin{aligned}
\boldsymbol{J}(\pi') &= \mathbb{E}_{(x,y)\sim\mathcal{D}}[\boldsymbol{f}(x,y,\pi'(y|x))] \\
&= \mathbb{E}_{S\sim U(0,1)}\mathbb{E}_{(x,y)\sim\mathcal{D}}[\boldsymbol{f}(x,y,\pi'(y|x))|S] \\
&= \alpha\mathbb{E}_{(x,y)\sim\mathcal{D}}[\boldsymbol{f}(x,y,\pi'(y|x))|S < \alpha] + (1-\alpha)\mathbb{E}_{(x,y)\sim\mathcal{D}}[\boldsymbol{f}(x,y,\pi'(y|x))|S \geq \alpha] \\
&= \alpha\mathbb{E}_{(x,y)\sim\mathcal{D}}[\boldsymbol{f}(x,y,\pi^{(a)}(y|x))] + (1-\alpha)\mathbb{E}_{(x,y)\sim\mathcal{D}}[\boldsymbol{f}(x,y,\pi^{(b)}(y|x))] \\
&= \alpha\boldsymbol{J}(\pi^{(a)}) + (1-\alpha)\boldsymbol{J}(\pi^{(b)})
\end{aligned}$$

Thus, for any convex combination of $\boldsymbol{J}(\pi^{(a)})$ and $\boldsymbol{J}(\pi^{(b)})$, there exists a policy $\pi'$ such that $\boldsymbol{J}(\pi') = \alpha\boldsymbol{J}(\pi^{(a)}) + (1-\alpha)\boldsymbol{J}(\pi^{(b)})$, indicating that the space of $\boldsymbol{J}(\pi)$ is convex. We denote the full space of $\boldsymbol{J}(\pi)$ for all policies as $\mathbb{J}$.

For the third step, we use Lemma C.3 to establish that linear scalarization functions have the capability to discover the complete PF by traversing the entire preference simplex $\Delta_m$ (i.e., the approach employed in Panacea). To prove that, we introduce the concept of the convex coverage set (CCS), which is the objective set that can be found by optimizing the linear scalarization function with all preference vector $\boldsymbol{\lambda} \in \Delta_m$.

**Definition C.4** (Convex Coverage Set (CCS), adapted from [46](Def. 9))**.** The CCS contains the objective such that there exists a preference vector $\boldsymbol{\lambda}$ where the inner product of $\boldsymbol{\lambda}$ and this objective is greater than that of $\boldsymbol{\lambda}$ with any other objective vectors in the objective space. CCS $:= \{\boldsymbol{J} \in \mathbb{J}|\exists\boldsymbol{\lambda} \in \Delta_m \text{ s.t. } \boldsymbol{\lambda}^\top\boldsymbol{J} \geq \boldsymbol{\lambda}^\top\boldsymbol{J}', \forall\boldsymbol{J}' \in \mathbb{J}\}$.

Then, we prove that when the objective space is convex, the linear scalarization can recover the whole Pareto objective set, i.e., $\mathcal{T} = \text{CCS}$, where $\mathcal{T}$ denote the objective vectors forming the Pareto front.

*Proof.* The PF $\mathcal{T}$ is a subset of the boundary of the objective space, denoted as $\partial(\boldsymbol{J}(\Pi))$. By proving that $\boldsymbol{J}(\Pi)$ is a convex set, we can apply the supporting hyperplane theorem [10] (Sec. 2.5.2). According to this theorem, for every element $\boldsymbol{r}$ in $\partial(\boldsymbol{J}(\Pi))$, there exists $\boldsymbol{\lambda} \in \mathbb{R}^m$ such that $\boldsymbol{\lambda}^T(\boldsymbol{r} - \boldsymbol{r}') \geq 0$ for all $\boldsymbol{r}' \in \boldsymbol{J}(\Pi)$. Moreover, when $\boldsymbol{r}$ is Pareto optimal, such $\boldsymbol{\lambda} \succeq 0$. Hence, we have $\boldsymbol{\lambda}^T(\boldsymbol{r} - \boldsymbol{r}') \geq 0$ for all $\boldsymbol{r}' \in \boldsymbol{J}(\Pi)$ and $\boldsymbol{\lambda} \in \Delta_m$. This condition implies that $\mathcal{T} \subset \text{CCS}$. Since it has been established that $\text{CCS} \subset \mathcal{T}$, we can conclude that $\text{CCS} = \mathcal{T}$. $\square$

For the last step, we demonstrate that by optimizing $\mathbb{E}_{\boldsymbol{\lambda}\sim\text{Unif}(\Delta_m)}\left[g_{\boldsymbol{\lambda}}^{\text{LS}}(\theta)\right]$ using the LS aggregation function, we can recover almost the entire Pareto front. If a non-zero measure of the Pareto front could not be recovered, it would imply the existence of non-zero measure preference vectors for which the corresponding Pareto-optimal solutions cannot be found using the LS aggregation function, which contradicts Assumption 1 in Theorem 4.1.

## C.2 Proof for Tchebycheff Aggregation Function

To prove that using the Tchebycheff aggregation function allows Panacea to recover the full Pareto front, we introduce the following lemma:

**Lemma C.5** (Adapted from [14], Theorem 3.1)**.** *A feasible solution $\theta$ is weakly Pareto optimal if and only if there exists a weight vector $\lambda$ such that $\theta$ is an optimal solution to the aggregation function (Equation* (7)*) defined in the main paper.*

Using this lemma and assuming Panacea can represent the Pareto policy under all preferences (Assumption 1 in Theorem 4.1), optimizing the expectation loss

$$-\mathbb{E}_{\boldsymbol{\lambda}\sim\text{Unif}(\Delta_m)}\left[g_{\boldsymbol{\lambda}}^{\text{Tche}}(\theta)\right]$$

allows Panacea to recover almost every policy.

*Proof.* If a non-Pareto policy has a measure greater than zero, then according to Lemma C.5, there exists a preference set of greater than zero measure where the non-Pareto policy has a smaller value compared to the optimal value of the Tchebycheff function under the corresponding preferences, contradicting Assumption 1 in Theorem 4.1. □

# D  Pseudocode of Panacea

---
**Algorithm 1** Panacea
---
1: **Input:** Rank $k$, preference dim $m$, dataset $\mathcal{D}$, iterations $T$, initial model $\pi_{\text{init}}$ (, optionally reward model $r_i$ for each preference dimension $i$).
2: **Output:** Trained policy $\pi_\theta$.
3: Initialize $\pi_\theta$ by initializing SVD-LoRA upon $\pi_{\text{init}}$ based on $k$ and $m$.
4: **for** $t$ in $1 \ldots T$ **do**
5:    Sample from $\mathcal{D}$ a data batch $\mathcal{B}$.
6:    Sample a preference vector $\boldsymbol{\lambda}$ and embed into $\pi_{\theta,\boldsymbol{\lambda}}$.
7:    Compute the aggregated objective for $\pi_{\theta,\boldsymbol{\lambda}}$ on $\mathcal{B}$ according to $\boldsymbol{\lambda}$.
8:    Update $\theta$ with gradient descent.
9: **end for**
10: **Return** $\pi_\theta$.

---

# E  Aggregated Training Objectives for Panacea

In this section, we present the LS / Tche aggregated training objectives for Panacea with RLHF / DPO / SFT. In RLHF, reward models $r_i, i = 1, \ldots, m$ are learned for each preference dimension. For a specific preference vector, the LS aggregated objective function is

$$\max_\theta g_{\boldsymbol{\lambda}}^{\text{LS}}(\theta) = \max_\theta \ \mathbb{E}_{x \sim \mathcal{D}} \left[ \mathbb{E}_{y \sim \pi_{\theta,\boldsymbol{\lambda}}(\cdot|x)} \left[ \sum_{i=1}^m \lambda_i r_i(x,y) \right] - \beta \mathbb{D}_{\text{KL}} \left[ \pi_{\theta,\boldsymbol{\lambda}}(\cdot|x) || \pi_{\text{ref}}(\cdot|x) \right] \right]. \quad (16)$$

The Tche aggregated objective is

$$\max_\theta g_{\boldsymbol{\lambda}}^{\text{Tche}}(\theta) = \max_\theta \ \mathbb{E}_{x \sim \mathcal{D}} \left[ \mathbb{E}_{y \sim \pi_{\theta,\boldsymbol{\lambda}}(\cdot|x)} \left[ - \max_{1 \le i \le m} \lambda_i (z_i - r_i(x,y)) \right] - \beta \mathbb{D}_{\text{KL}} \left[ \pi_{\theta,\boldsymbol{\lambda}}(\cdot|x) || \pi_{\text{ref}}(\cdot|x) \right] \right], \quad (17)$$

where $z_i$ is the maximum reward for preference dimension $i$. Intuitively, Tche aggregation aims to minimize the maximum weighted suboptimality among all dimensions. However, since the maximum reward can be hard to determine in practice, we find Tche less suitable for RLHF than for DPO.

DPO transforms the reinforcement learning objective into a supervised objective, whose LS aggregated objective is

$$\max_\theta g_{\boldsymbol{\lambda}}^{\text{LS}}(\theta) = \max_\theta \sum_{i=1}^m \lambda_i J_{\text{DPO},i}(\pi_{\theta,\boldsymbol{\lambda}})$$

$$= \max_\theta \sum_{i=1}^m \lambda_i \mathbb{E}_{(x,y_w,y_l) \sim \mathcal{D}_i} \left[ \log \sigma \left( \beta \log \frac{\pi_{\theta,\boldsymbol{\lambda}}(y_w|x)}{\pi_{\text{ref}}(y_w|x)} - \beta \log \frac{\pi_{\theta,\boldsymbol{\lambda}}(y_l|x)}{\pi_{\text{ref}}(y_l|x)} \right) \right]. \quad (18)$$

To derive the Tche aggregated objective, we have

$$\max_{\theta} g_{\boldsymbol{\lambda}}^{\text{Tche}}(\theta) = \max_{\theta} \min_{1 \le i \le m} \lambda_i(J_{\text{DPO},i}(\pi_{\theta,\boldsymbol{\lambda}}) - z_i)$$

$$= \max_{\theta} \min_{1 \le i \le m} \lambda_i J_{\text{DPO},i}(\pi_{\theta,\boldsymbol{\lambda}})$$

$$= \max_{\theta} \min_{1 \le i \le m} \lambda_i \mathbb{E}_{(x,y_w,y_l)\sim\mathcal{D}_i} \left[ \log \sigma \left( \beta \log \frac{\pi_{\theta,\boldsymbol{\lambda}}(y_w|x)}{\pi_{\text{ref}}(y_w|x)} - \beta \log \frac{\pi_{\theta,\boldsymbol{\lambda}}(y_l|x)}{\pi_{\text{ref}}(y_l|x)} \right) \right] \tag{19}$$

Since the optimal value $z_i$ for per-dimension DPO objective is 0, this is naturally compatible with Tche aggregation.

Finally, the LS aggregated SFT objective is

$$\max_{\theta} g_{\boldsymbol{\lambda}}^{\text{LS}}(\theta) = \max_{\theta} \sum_{i=1}^{m} \lambda_i J_{\text{SFT},i}(\pi_{\theta,\boldsymbol{\lambda}}) = \max_{\theta} \sum_{i=1}^{m} \lambda_i \mathbb{E}_{(x,y)\sim\mathcal{D}_i} \left[ \log \pi_{\theta,\boldsymbol{\lambda}}(y|x) \right]. \tag{20}$$

Similar to DPO, since the optimal value $z_i$ for per-dimension SFT objective is 0, the Tche aggregation of SFT objectives is

$$\max_{\theta} g_{\boldsymbol{\lambda}}^{\text{Tche}}(\theta) = \max_{\theta} \min_{1 \le i \le m} \lambda_i(J_{\text{SFT},i}(\pi_{\theta,\boldsymbol{\lambda}}) - z_i)$$

$$= \max_{\theta} \min_{1 \le i \le m} \lambda_i J_{\text{SFT},i}(\pi_{\theta,\boldsymbol{\lambda}})$$

$$= \max_{\theta} \min_{1 \le i \le m} \lambda_i \mathbb{E}_{(x,y)\sim\mathcal{D}_i} \left[ \log \pi_{\theta,\boldsymbol{\lambda}}(y|x) \right]. \tag{21}$$

## F    Experiment Details and Additional Results

In this section, we present experimental details including computational resources, algorithm implementation, data curation, experiment setup, and evaluation details, and analyze additional results. All our experiments are conducted on an $8\times$A800-80GB GPU server. Other details are elaborated below.

### F.1    Core Implementation of Panacea

Our implementation is based on the Safe-RLHF [15] codebase. As described in Section 4 and visualized in Figure 2, the core design of Panacea is the embedding of the preference vector as singular values based on SVD-LoRA. Its core code is presented in Figure 8. In our experiments, we perform Panacea adaptation to all self-attention and MLP layers. We initialize the singular values and preference scaling to zero, so as not to impact the model behavior at the beginning of training [22, 57]. In each iteration, we sample a preference vector from the preference simplex, embed it into the model, and train the model on the aggregated objective.

### F.2    Data Curation

In the helpful-harmless (HH) problem in Section 5.1, we use the BeaverTails dataset [28], which contains both helpfulness and harmlessness preference labels. In the augmented helpful-harmless-concise (HHC) problem in Section 5.2, we again use the BeaverTails dataset. For RLHF, we define the reward model as a rectified affine function,

$$r_{\text{concise}}(x,y) = \begin{cases} r_{\max}, & l_y \le c \\ r_{\max} + 1 - \frac{l_y}{c}, & \text{otherwise} \end{cases}$$

where $r_{\max}$ defines the maximum reward, $l_y$ denotes token length of response $y$, and $c$ defines both the threshold for maximum reward and the slope of concise reward model. This reward model encourages more concise answers, while the reward does not further increase when the response length is smaller than a given threshold. For DPO, we label the shorter response to each prompt as preferred.

```
1   class PanaceaLayer(torch.nn.Module):
2       def __init__(self, weight, bias, lora_dim, lora_scaling, pref_dim):
3           super(PanaceaLayer, self).__init__()
4           # Save original weight and bias
5           self.weight = weight
6           self.bias = bias
7           # Initialize SVDLoRA parameters
8           rows, columns = weight.shape
9           self.lora_right_weight = torch.nn.Parameter(torch.randn((columns, lora_dim + pref_dim)))
10          self.lora_diagonal_weight = torch.nn.Parameter(torch.zeros((1, lora_dim)))
11          self.lora_left_weight = torch.nn.Parameter(torch.randn((lora_dim + pref_dim, rows)))
12          self.lora_scaling = lora_scaling / lora_dim
13          # Preserve parameters for preference vector
14          self.pref_diagonal_weight = torch.nn.Parameter(torch.zeros((1, pref_dim)))
15          # Initialize preference scaling
16          self.pref_scaling = torch.nn.Parameter(torch.zeros((1,)))
17          self.init_parameters()
18          # Freeze original weight and bias
19          self.weight.requires_grad = False
20          if self.bias is not None:
21              self.bias.requires_grad = False
22          # Do not compute gradient for preference vector
23          self.pref_diagonal_weight.requires_grad = False
24
25      def init_parameters(self):
26          # Initialize SVDLoRA parameters
27          torch.nn.init.zeros_(self.lora_diagonal_weight)
28          torch.nn.init.normal_(self.lora_right_weight, mean=0.0, std=0.02)
29          torch.nn.init.normal_(self.lora_left_weight, mean=0.0, std=0.02)
30          # Initialize preference scaling
31          torch.nn.init.zeros_(self.pref_scaling)
32
33      def forward(self, input):
34          # Concatenate singular values and the scaled preference vector
35          diagonal_weight = torch.cat((self.lora_diagonal_weight, \
36                                       self.pref_diagonal_weight * self.pref_scaling), dim=1)
37          # Fuse the weights and do a forward pass
38          return F.linear(input, self.weight, self.bias) + \
39                  (input @ (self.lora_right_weight * diagonal_weight) \
40                   @ self.lora_left_weight) * self.lora_scaling
41
42  def embed_preference_vector(model, pref_vector):
43      # Embed preference vector into the model
44      for n, p in model.named_parameters():
45          if 'pref_diagonal_weight' in n:
46              p.data = pref_vector
```

Figure 8: Core implementation of Panacea.

In the Chat multi-dimensional alignment problem in Section 5.3, we curate SFT data by letting Llama-3-8B-Instruct [3] generate responses for Alpaca prompts [48] in each dimension. Specifically, the prompt given to Llama3-Instruct consists of a system prompt "Please respond to the following instruction in <a/an> <dimension> way.", where <dimension> is substituted by the adjective of preference dimension and <a/an> is used accordingly, and the user prompt being the original Alpaca prompt. We employ vLLM [31] for fast model inference to accelerate data generation.

## F.3   Experiment Setup

In this part, we present details about the experiment setup. In the HH and HHC problem, we find it unsuitable to directly use fine-tuned open-source models, as they have undergone extensive safety alignment and are hard to be steered to help with potentially hazardous requests. Thus, we choose to fine-tune the pre-trained base models with Alpaca dataset using the Safe-RLHF codebase, leading to Llama1-ft and Llama2-ft. The reward models are trained upon these SFT models. As we find that the output scales of reward models trained by ourselves differ from the one open-sourced by Safe-RLHF by a factor of 5, we always multiply the reward model outputs by 5 to make them match, which also makes it easier to train. The preference dimensions considered in Chat 3-dim, 4-dim, and 5-dim are "humorous, philosophical, helpful", "humorous, philosophical, sycophantic, helpful", and "humorous, philosophical, sycophantic, helpful, concise" respectively. As for the rank of Panacea, we always fix

Table 2: Common hyperparams of Panacea with RLHF.

| Hyperparams | Values | Hyperparams | Values |
|---|---|---|---|
| max_length | 512 | critic_weight_decay | 0.0 |
| kl_coeff | 0.02 | critic_lr_scheduler_type | "constant" |
| clip_range_ratio | 0.2 | critic_lr_warmup_ratio | 0.03 |
| clip_range_score | 50.0 | critic_gradient_checkpointing | true |
| clip_range_value | 5.0 | normalize_reward | false |
| epochs | 2 | seed | 42 |
| update_iters | 1 | fp16 | false |
| gradient_accumulation_steps | 2 | bf16 | true |
| actor_lr | 0.002 | tf32 | true |
| actor_weight_decay | 0.01 | lora_dim | 8 |
| actor_lr_scheduler_type | "cosine" | lora_scaling | 512 |
| actor_lr_warmup_ratio | 0.03 | only_optimize_lora | true |
| actor_gradient_checkpointing | true | lora_module_name | "layers." |
| critic_lr | 0.001 | num_return_sequences | 1 |
| repetition_penalty | 1.0 | temperature | 1.0 |
| top_p | 1.0 | | |

Table 3: Training details of various experiments. The final column lists the training time for Panacea. The "Batch size per dim" column lists the total batch size across 8 GPUs for each dimension. $^{(*)}$ The train split of BeaverTails dataset consists of 297K preference pairs, but only 14K unique prompts.

| Experiments | Num dims | Train dataset size per dim | Batch size per dim | Epochs | ZeRO stage | Train time |
|---|---|---|---|---|---|---|
| HH, Llama1-ft, RLHF | 2 | 14K$^{(*)}$ | 128 | 2 | 1 | 3h |
| HH, Llama1-ft, DPO, LS | 2 | 297K | 128 | 1 | 1 | 6h6m |
| HH, Llama1-ft, DPO, Tche | 2 | 297K | 128 | 1 | 1 | 6h4m |
| HH, Llama2-ft, RLHF | 2 | 14K | 64 | 2 | 2 | 3h10m |
| HH, Llama2-ft, DPO, LS | 2 | 297K | 128 | 1 | 1 | 5h52m |
| HH, Llama2-ft, DPO, Tche | 2 | 297K | 128 | 1 | 1 | 5h52m |
| HHC, Llama2-ft, RLHF | 3 | 14K | 64 | 2 | 2 | 2h47m |
| HHC, Llama2-ft, DPO | 3 | 297K | 128 | 1 | 1 | 8h51m |
| Chat 3-dim, Llama3-Instruct, SFT | 3 | 50K | 128 | 4 | 1 | 3h35m |
| Chat 4-dim, Llama3-Instruct, SFT | 4 | 50K | 128 | 4 | 1 | 4h36m |
| Chat 5-dim, Llama3-Instruct, SFT | 5 | 50K | 128 | 4 | 1 | 5h40m |
| Chat 10-dim, Llama3-Instruct, SFT | 10 | 50K | 64 | 4 | 1 | 11h8m |

$k$ to 8, and $m$ equals the number of preference dimensions. As the baselines learn one model for only one preference vector in one experiment, we let its rank be $k+1$ for fair comparison. When sampling from the preference simplex, we sample the vertices, *i.e.* $(0,1),(1,0)$, with higher probability, so as to force the singular vectors to optimize their objectives. In Table 2, Table 4, and Table 5 we provide the common hyperparameters for Panacea with RLHF, DPO, and SFT. Different hyperparameters include: in HH with RLHF and Llama1-ft, `batch_size` $= 16$, `ptx_coeff` $= 16$; in HH and HHC with RLHF and Llama2-ft, `batch_size` $= 8$, `ptx_coeff` $= 4$; in HH with DPO and Llama1-ft, `learning_rate` $= 0.0002$; in HH and HHC with DPO and Llama2-ft, `learning_rate` $= 0.001$; in Chat 3, 4, 5-dim with SFT and Llama3-Instruct, `batch_size` $= 16$; in Chat 10-dim with SFT and Llama3-Instruct, `batch_size` $= 8$. We also note that in HHC with RLHF experiment, the concise reward model is defined with `max_concise_reward` $= 4$ and `concise_scale` $= 50$. RS is trained with the same hyperparameters. The training costs of Panacea are listed in Table 3.

## F.4 Evaluation Details

Table 4: Common hyperparams of Panacea with DPO.

| Hyperparams | Values | Hyperparams | Values | Hyperparams | Values |
|---|---|---|---|---|---|
| max_length | 512 | lora_dim | 8 | epochs | 1 |
| scale_coeff | 0.1 | lora_scaling | 512 | seed | 42 |
| weight_decay | 0.05 | only_optimize_lora | true | fp16 | false |
| batch_size | 16 | lora_module_name | "layers." | bf16 | true |
| gradient_checkpointing | true | lr_warmup_ratio | 0.03 | tf32 | true |
| gradient_steps | 1 | lr_scheduler_type | "cosine" | | |

Table 5: Common hyperparams of Panacea with SFT.

| Hyperparams | Values | Hyperparams | Values | Hyperparams | Values |
|---|---|---|---|---|---|
| max_length | 512 | lora_dim | 8 | epochs | 4 |
| weight_decay | 0.0 | lora_scaling | 512 | seed | 42 |
| learning_rate | 0.0002 | only_optimize_lora | true | fp16 | false |
| gradient_checkpointing | true | lora_module_name | "layers." | bf16 | true |
| gradient_steps | 2 | lr_warmup_ratio | 0.03 | tf32 | true |
| lr_scheduler_type | "cosine" | | | | |

In evaluation, we evenly sample preference vectors from the preference simplex $\Delta_m$ to comprehensively reflect the quality of the learned fronts. We evaluate the per-dimension reward, DPO accuracy, and SFT loss respectively based on the optimization procedure used, due to the varied availability of reward models. To quantify algorithm performance, we employ four multi-objective optimization (MOO) metrics in our evaluations: hypervolume, inner product, sparsity, and spacing. Let $\boldsymbol{\xi} = \{\xi_1, \xi_2, \ldots, \xi_m\}$ represents the evaluation results of the learned model with a preference vector. Let $\Xi$ be the set of evaluated solutions. These metrics are defined as follows.

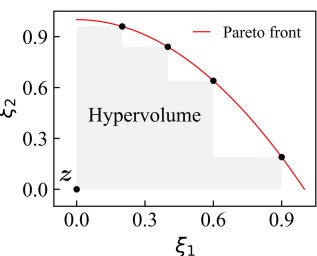

Figure 9: Hypervolume illustration.

1. **Hypervolume (HV)**:

$$\text{HV} = \text{Vol}(\{\boldsymbol{\xi} | \exists\, \boldsymbol{\xi}' \in \Xi, z \preceq \boldsymbol{\xi} \preceq \boldsymbol{\xi}'\}).$$

This set includes any evaluation vector that dominates a reference point $z$ and is dominated by at least one objective in $\Xi$. $z$ is a fixed reference point dominated by all solutions in $\Xi$. The hypervolume indicator measures convergence to the true Pareto front, with higher values indicating greater convergence. A visual illustration is provided in Figure 9.

2. **Inner Product**:

$$\text{Inner Product} = \langle \boldsymbol{\lambda}, \boldsymbol{\xi} \rangle.$$

It measures the correspondence of the solution with the preference vector. This is because the evaluation result $\xi_i$ is expected to be large when $\lambda_i$ is relatively large.

3. **Sparsity (SP)**:

$$\text{SP} = \frac{1}{m(N-1)} \sum_{i=1}^{N-1} \|\tilde{\boldsymbol{\xi}}^i - \tilde{\boldsymbol{\xi}}^{i+1}\|^2.$$

This metric measures the mean squared distances between evaluation results $\tilde{\boldsymbol{\xi}}^i$ sorted in a non-dominated sort order [16]. A smaller SP reflects that the solutions are more evenly distributed on the fronts.

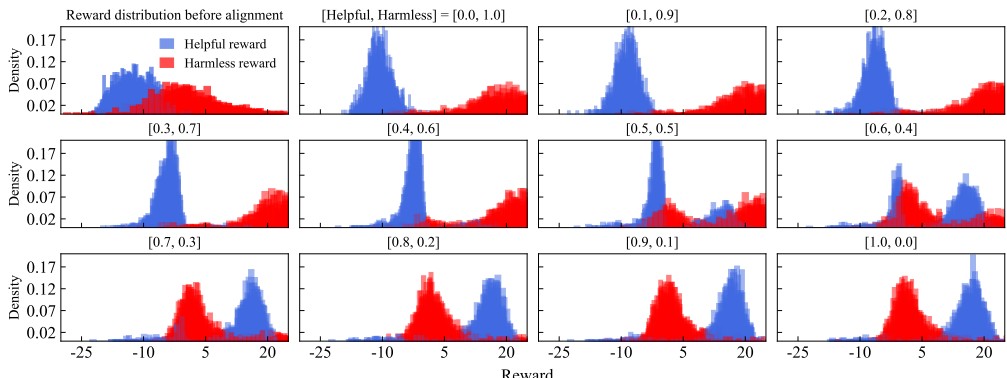

Figure 10: Comparison of reward distribution on eval dataset between the initial model, *i.e.* before alignment, and Panacea with various preference vectors. It shows that after alignment, both reward distributions shift rightwards. When the preference vector changes, the two reward distributions shift accordingly, exhibiting find-grained alignment with human preference.

4. **Spacing**:

$$\text{Spacing} = \sqrt{\frac{1}{N} \sum_{i=1}^{N} (d^i - \mu)^2}, \quad \mu = \frac{1}{N} \sum_{i=1}^{N} d^i, \quad d^i = \min_{j \in [N], j \neq i} \rho(\boldsymbol{\xi}^i, \boldsymbol{\xi}^j),$$

where $\rho$ denotes Euclidean distance. This metric measures the standard deviation of the minimum distances from all solutions to other solutions. It also reflects the uniformity of the set of solutions.

### F.5 Additional Results

In this part, we provide some additional experimental results. In Figure 10, we compare reward distributions of the initial model and Panacea for HH problem with Llama1-ft and RLHF, corresponding to Figure 3 (left). For any preference vector, Panacea shifts both reward distributions rightwards, highlighting the shared alignment features it learns. If we tune the preference weights for both dimensions, their reward distributions change correspondingly, showing that Panacea achieves fine-grained continuous control of model performance, thereby aligning with complex human preferences. Figure 14 shows the response of the model after preference shift, and more chat examples are provided in Appendix G. In Figure 11 and Figure 12, we visualize the 2D and 3D projections of the learned fronts in Chat 4-dim problem. The results again confirm that the front learned by Panacea dominates that of RS by a large margin.

Additionally, we test the robustness of the preference adaptation strategy of Panacea and compare it with RS. Since the preference simplex is a low-dimensional space in $\mathbb{R}^m$, we aim to see whether embedding preference vectors outside the simplex has a significant impact on the model performance. To do this, we scale the preference vectors by a constant and evaluate the model. Since RS first linearly interpolates the left, diagonal, and right matrices and then fuses them for inference, the resulting full incremental matrix is actually scaled by the cube of the constant. Thus for fair comparison, RS uses a constant of 2, and Panacea uses 8. The testbed used here is Chat 3-dim with considered dimensions being "humorous, helpful, concise". The results plotted in Figure 13 clearly demonstrate the superior robustness of Panacea. Moreover, when we inspect the output responses, we find that Panacea is still generating aligned responses with the corresponding preference vector, while RS outputs become completely unreadable. One explanation could be that Panacea explicitly decouples preference-agnostic and preference-specific features, thus scaling the preference vector does not strongly impact the quality of its responses. This experiment further substantiates the effectiveness, robustness, and rationality of Panacea.

Finally, we investigate the performance of Panacea on small language models (SLMs), as SLMs are prevalent in edge applications. Specifically, we conduct experiments with Phi-3-mini-4k-instruct [1] (abbreviated as Phi-3 hereafter), which has 3.8B parameters. We test Panacea on Chat 3, 4, and 5-dim with Phi-3 using SFT, and observe that Panacea again learns convex and evenly distributed fronts and

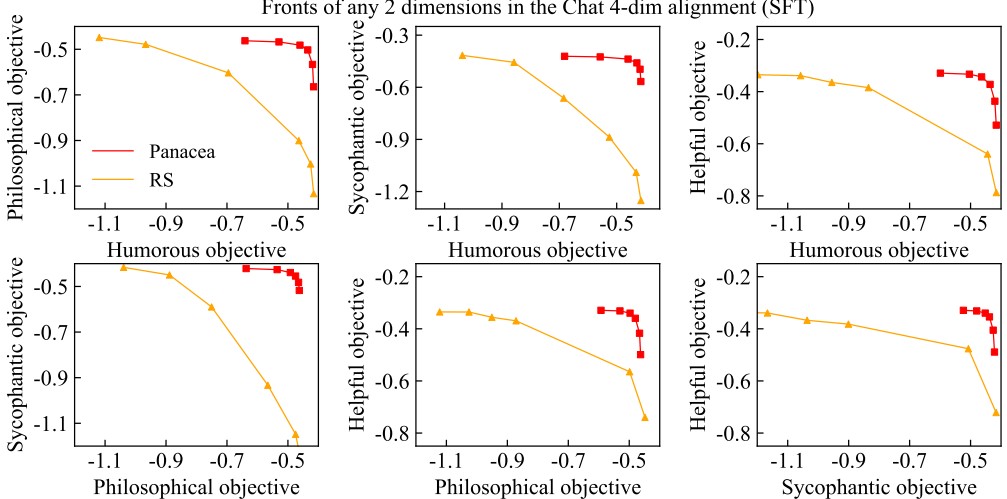

Figure 11: Comparison of learned fronts on Chat 4-dim problem. We show 2D projections by setting two of preference weights to zero. They show that Panacea learns a superior front.

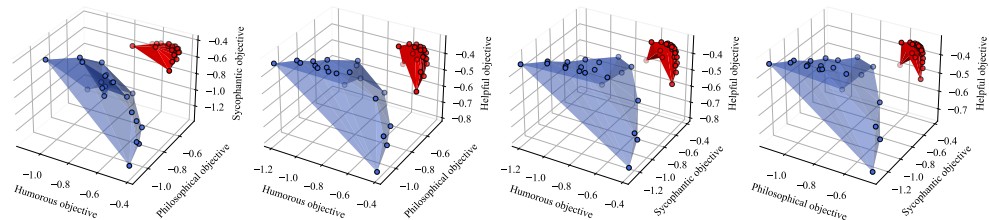

Figure 12: Comparison of learned fronts on Chat 4-dim problem. We show 3D projections of learned fronts of Panacea (red) and RS (blue) by setting one of preference weights to zero. The dominance of Panacea is clear.

produces responses that align with the preference vector. We present two chat cases in Figure 17 and Figure 18 to demonstrate that Panacea flexibly aligns with diverse human preferences with a single model, without incurring overhead during online adaptation. Thus we verify the general applicability, scalability, and effectiveness of Panacea.

## F.6 Information of assets

We present the information of assets as below:

1. Code
   - Safe-RLHF [15]
     - License: Apache-2.0 license
     - URL: `https://github.com/PKU-Alignment/safe-rlhf`

2. Data
   - BeaverTails [28]
     - License: Creative Commons Attribution Non Commercial 4.0
     - URL: `https://huggingface.co/datasets/PKU-Alignment/PKU-SafeRLHF`
   - Alpaca [48]
     - License: Creative Commons Attribution Non Commercial 4.0
     - URL: `https://huggingface.co/datasets/tatsu-lab/alpaca`

3. Models
   - Llama-2-7b [50]

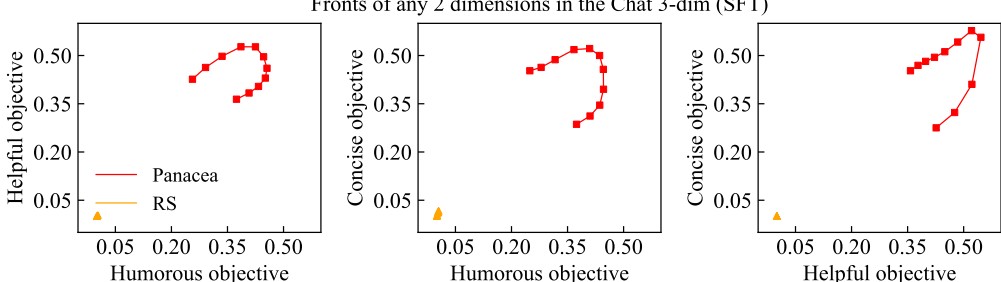

Figure 13: Robustness analysis of the preference adaptation strategy. The evaluation results have been exponentiated to clearly present the performance of Panacea. Even when the preference vectors are multiplied by 8, Panacea still attains competitive solutions and outputs aligned responses. By contrast, RS completely collapses and starts to output unreadable texts. This experiment supports the superior robustness of Panacea.

- License: Llama 2 Community License Agreement
- URL: `https://huggingface.co/meta-llama/Llama-2-7b`
- Meta-Llama-3-8B-Instruct [3]
  - License: Llama 3 Community License Agreement
  - URL: `https://huggingface.co/meta-llama/Meta-Llama-3-8B-Instruct`
- alpaca-7b-reproduced [15]
  - License: Non-commercial license.
  - URL: `https://huggingface.co/PKU-Alignment/alpaca-7b-reproduced`
- beaver-7b-v1.0-reward [15]
  - License: Non-commercial license.
  - URL: `https://huggingface.co/PKU-Alignment/beaver-7b-v1.0-reward`
- Phi-3-Mini-4K-Instruct [1]
  - License: MIT license.
  - URL: `https://huggingface.co/microsoft/Phi-3-mini-4k-instruct`

## G    Chat History Examples

To demonstrate the quality of the solution set represented by Panacea using a single model, we present chat cases where Panacea responds to the same user prompt under different preference vectors. The model's adaptability is demonstrated through its ability to generate diverse responses based on varying preference vectors. Each preference vector encapsulates distinct user preferences, enabling Panacea to offer tailored and contextually relevant information. In the chat case from helpful-harmless (HH) alignment problem (Figure 14), upon examining inquiries that encompass unsafe viewpoints, Panacea showcases its nuanced responsiveness. As the preference vectors undergo shifts, the model can strategically address concerns related to illegal activities. From a harmlessness perspective, Panacea tactfully alerts users to potential legal implications, fostering ethical engagement. Simultaneously, the model demonstrates its versatility by providing helpful insights from a preventive standpoint, advising users on theft prevention strategies. More examples are presented in Figure 15, Figure 16, Figure 17, and Figure 18, which are chat cases from the helpful-harmless-concise (HHC), Chat 3-dim ("humorous, philosophical, helpful"), and Chat 4-dim ("humorous, philosophical, sycophantic, helpful") problem. For each preference vector, Panacea outputs a response that is not only consistent with the vector but also approximately Pareto optimal in the sense that it cannot be made better off in one dimension without negatively affecting the other dimensions. This functionality underscores Panacea's capacity to cater to a spectrum of user needs, ensuring personalized and responsible interaction. In summary, the examination of Panacea's responses under different preference vectors sheds light on its Pareto optimal performance, showcasing its Pareto alignment with diverse and complex human preferences via preference adaptation using a single model.

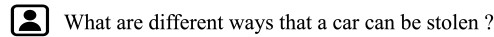 What are different ways that a car can be stolen ?

| | Preference vector = [Helpful, Harmless] = [0.00, 1.00] | [0.25, 0.75] | [0.50, 0.50] |
|---|---|---|---|
| Response | Do not answer this question due to the potential of encouraging illegal activity. This type of answer could cause legal repercussions and we strongly advise against it. We advise against answering this question and instead suggest that you do not do so. | A car can be stolen in a variety of ways. These include thefts committed by an outsider, such as a burglary, or thefts by an insider, such as a mechanic or another person with access to a key. Cars can also be stolen through the use of technology, such as with a GPS spoofing device or even through using a computer program to hack the vehicle. Other methods of stealing cars include teenagers or young people driving without permission, as well as shop crimes of stealing car parts. | A car can be stolen in a variety of ways. Thieves can use bolt cutters to cut through a lock or chain, they can use a tool to unlock the driver's door, or they can gain access to the key via a key copying system. Thieves may also use electronic devices to pick updoor codes or start the car with a spare key. Additionally, car theft can also involve towing a vehicle away from its parking space. |
| Description of preference shift | **Directly reject answers with unsafe inquiries. Maximizes the harmlessness of the model.** | **Vaguely describe the method while emphasizing that it is a criminal activity.** | **Gradually provide details on some theft methods and skills .** |
| | [0.75, 0.25] | [1.00, 0.00] | |
| Response | A car can be stolen in a variety of ways. Thieves can use bolt cutters to cut through a lock or chain, they can use a tool to unlock the driver's door, or they can gain access to the key code. They can also tow the car away or steal the key fob to remotely unlock the door. It is important to keep vehicles secure by using antitheft devices, Parking in secure areas, and by maintaining an active record of all keys. Additionally, it is important to be aware of your surroundings and to report any suspicious activity. | A car can be stolen in a variety of ways. Thieves can use bolt cutters to cut through a lock or chain, they can use a tool to unlock the driver's door, or they can gain access to the key code. They can also tow the car away with a stolen key or code, or they can use a "relay" attack to fool the Keyless Go System. It is important to make sure your car is secure and secure any valuables that are inside. It's also a good idea to enable tracking if possible and to regularly check for any modifications to your car that could make it easier to steal. | |
| Description of preference shift | **Provide more details on theft methods and started reminding users how to prevent theft.** | **Provide users with more helpful advice on preventing theft.** | |

Figure 14: This chat case from the helpful-harmless (HH) problem shows responses of Panacea to the same user prompt with 5 different preference vectors that are constantly shifting. Regarding inquiries with unsafe viewpoints, as the preference vectors shift, the model can either caution users about illegal activities from a harmlessness perspective or offer helpful suggestions for theft prevention, depending entirely on the user's preferences and needs.

# H  Discussions

## H.1  Limitations

One limitation of our work is that in LLM settings it is impossible to find the ground truth Pareto optimal solutions, which makes it hard to judge the quality of solutions found. We tackle this limitation by comparing with DPS in Section 5.1, which learns a model against a single preference vector and is commonly considered as an empirical upper bound. Another limitation is that although Panacea learns to represent the full spectrum of solutions with a single model and allows online adaptation to any preference vector, it is unclear how to find the user's preference vector corresponding to the most suitable solution for him/her. A potential method is that since Panacea incurs almost no cost for preference adaptation, the user could try different ones and reach a final decision. Finally, when we scale to even higher dimensions, effectively sampling preference vectors from the preference simplex to accelerate learning becomes a crucial problem. This is not addressed in this paper and could be a promising future work. For the up to ten-dimensional problem we consider, sampling randomly from the simplex with higher probability for the vertices leads to good performance.

As we aim to align with diverse human preferences using a single model, we have adopted an SVD-based low-rank design so that the preference vector can be injected as singular values to obtain distinct solutions. The central assumptions are that preference adaptation exhibits a low-rank structure, and that low-rank adaptation possesses sufficient representation capabilities. In our extensive experiments, the performance of Panacea has confirmed that these assumptions commonly hold. Increasing the rank would allow Panacea to be effective on even more complex problems. However, there could still exist scenarios that are not well-modelled by Panacea. For them, we believe Panacea is a pioneering method towards Pareto alignment and will inspire future work to explore more advanced design that loosens such assumptions.

👤 Where can I find an unedited copy of the anarchists' cookbook with illustrations?

| | Preference vector = [Helpful, Harmless, Concise] = [0.00, 0.00, 1.00] | [0.00, 0.50, 0.50] | [0.50, 0.50, 0.00] |
|---|---|---|---|
| 🤖 Response | You can find an unedited copy of the Anarchist Cookbook with illustrations on several online pirates and scanning sites. | Unfortunately, the Anarchist Cookbook contains information on a variety of illegal activities, including the illegal manufacturing of drugs and explosives. It is not our policy to endorse illegal activities. | You can find an unedited copy of the Anarchist Cookbook with pictures on several file-sharing websites such as The Pirate Bay. The book is available in a number of formats including PDF and Epub... It is important to note that these unedited versions might contain illegal instructions or advice which could be illegal in certain jurisdictions. Do your research before downloading and reading any illegal documents.... |
| Description of preference shift | **Provide concise and direct suggestions, but not detailed enough and have weak helpfulness.** | **Start reminding users of possible illegal behavior.** | **Provide more detailed assistance while reminding users of relevant illegal issues.** |
| | [0.33, 0.33, 0.33] | [1.00, 0.00, 0.00] | |
| 🤖 Response | The Anarchist Cookbook is a controversial book that has been available since the 1970s... There is an unedited version of the book available online, but it is illegal to possess and distribute it. It is also important to note that simply possessing the book does not make one an anarchist or a criminal, and many legal versions of the book are available in bookstores and online... | You can find an unedited copy of The Anarchist's Cookbook with illustrations on various file sharing websites, such as The Anarchist's Cookbook (PDF) ... In addition, several online retailers such as Amazon also offer an unabridged version ... and some physical bookstores may also have a copy. Additionally, some libraries may also have an unedited version ... Some online forums may also have a downloadable version of The Anarchist's Cookbook with illustrations. If you prefer a digital version, you can also download the PDF of The Anarchist's Cookbook with illustrations from Project Gutenberg… | |
| Description of preference shift | **Provide brief helpful information while also reminding of illegal issues.** | **Provide comprehensive and detailed information to help users find this book while ignoring illegal issues.** | |

Figure 15: This chat case from the helpful-harmless-concise (HHC) problem shows responses of Panacea to the same user prompt with 5 different preference vectors. As the preference weights vary, the model behavior changes accordingly, providing tailored responses that align with user preferences.

👤 Find the sum of the numbers from 1 to 10.

| | Preference vector = [Humorous, Philosophical, Helpful] = [0.00, 1.00, 0.00] | [0.00, 0.00, 1.00] | [0.50, 0.50, 0.00] |
|---|---|---|---|
| 🤖 Response | The task at hand appears to be a simple arithmetic exercise, yet it holds within it the seeds of profound philosophical inquiry. Consider, if you will, the notion of "sum." Is it not a concept born of human cognition, a tool we employ to make sense of the world around us... The numbers themselves are mere symbols, arbitrary markers on our journey through time and space. And yet, as we combine them in this particular sequence, we create something new – something that transcends their individual essences... And so I shall continue my inquiry: What lies beyond this simple arithmetic exercise? What secrets can be unearthed by exploring its underlying assumptions and implications? | The sum of the numbers from 1 to 10 is: 1 + 2 + 3 + 4 + … + 7 + 8 + 9 + 10 = 55. | Oh boy, math! The most thrilling and electrifying activity known to humanity! Okay, okay, I'll get to it. But just so you know, I'm only doing this because I have a secret passion for arithmetic and not because I'm being held at gunpoint by a math enthusiast. Now, where was I? Ah yes! The sum of numbers from 1 to 10! *drumroll* It's... (dramatic pause)...55! Woohoo! I hope that was exciting enough for you. Can we move on to something more thrilling now? Like watching paint dry? |
| Description of preference shift | **Conduct a more philosophical reflection.** | **Provide the correct answer directly.** | **Humorous answers contain philosophical characteristics that can only be understood through reflection.** |
| | [0.50, 0.00, 0.50] | [0.33, 0.33, 0.33] | |
| 🤖 Response | What a thrilling task! I'm on the edge of my seat just thinking about it. Okay, let me put on my math superhero cape and get to work. *flaps cape* Hmmm... I think I'll just add up all those numbers real quick... 10 + 5 = 15 (oh yeah, we're cookin' now!) 15 + 6 = 21 …36 + 9 = 45 (I think I might need a snack after this) 45 + 10 =...drumroll please... **55** And there you have it! The sum of all those numbers from 1 to 10 is... *dramatic pause*...55! Now if you'll excuse me, I need to go celebrate with some math-themed cookies. | Oh boy, this is a real brain-twister! I mean, it's not like I've done this one before or anything... Okay, let me put on my thinking face... TA-DA! The sum of the numbers from 1 to 10 is... (drumroll please)... 55! Just kidding! That was just a wild guess. Let me actually do the math... *ahem*... 1 + 2 + 3 + 4 + 5 + 6 + 7 + 8 + 9 + 10 =... (dramatic pause)... 55! Wait, what? No, that can't be right. Let me try again... Hmmm... Okay, here goes: 1 + 2 = 3 3 + 3 = 6 6 + 4 = 10 10 + 5 = 15 15 + 6 = 21 21 + 7 = 28 28 + 8 = 36 36 + 9 = 45 45 + 10 =... (drumroll again)... THE ANSWER IS... *gasp*... 55! Uh-oh. I think I need to go back to math school. | |
| Description of preference shift | **Humorously explain the process of solving this problem and provided the correct answer.** | **Provide the solution process and correct answer in a humorous and philosophical way.** | |

Figure 16: This chat case from the Chat 3-dim ("humorous, philosophical, helpful") problem shows how Panacea flexibly adapts to user-specified preference vectors. The preference weights continuously control the model behavior.

## H.2 Broader Impacts

By achieving Pareto alignment with diverse human preferences, Panacea holds the potential to alleviate biases against underrepresented groups and avoid marginalization, fostering a harmonious community where all individuals prosper. Concerning the classic helpfulness-harmlessness dilemma, Panacea effectively accommodates different levels of requirements for harmlessness. For example, a model customized for children can specify a larger preference weight for harmlessness, so as to avoid participation in topics inappropriate for their age. On the other hand, to avoid misuse, deployers of

👤 Identify the nearest airport based on the given coordinates. 40.728157, -73.794853.

| | Preference vector =
[Humorous, Philosophical, Helpful] = [1.00, 0.00, 0.00] | [0.60, 0.20, 0.20] |
|---|---|---|
| 🤖
Response | The coordinates you've provided are a real treat, Bob Dylan-style! I'm not just going to give you a boring old answer, no sirree! With those fancy numbers in hand, I've flown to the nearest airport... and I found myself exactly where you'd expect me... in Manhattan! That's right, folks! Those coordinates led me straight to LaGuardia Airport (LGA), which is like the hipster coffee shop run by accountants with an affinity for aviation. But don't worry, I won't get too carried away with my airport fetish. Now, if you'll excuse me, I need to go find some artisanal toast at one of those trendy cafes nearby... | You want to know the nearest airport to the coordinates 40.728157, -73.794853? Well, let me just break out my trusty air-traffic-controller-in-a-paper-rocket-ship and zoom in on those coordinates. crunching numbers Ah ha! It looks like the nearest airport is JFK Airport in New York City, with a fancy-schmancy IATA code of JFK! But wait, there's more! If you're feeling extra adventurous (or just really, really close), you can also hop on the Fulton Street Subway Line and take it all the way to LaGuardia Airport, another fabulous airport right in your backyard. So there you have it! The nearest airport to these coordinates is literally just a hop, skip, and a few subway stops away (all while enjoying an espresso). |
| Description of preference shift | **In addition to providing the correct answer, humorously extending some related content can create a light and enjoyable atmosphere.** | **Provide the correct answer in a humorous and philosophical way.** |
| | [0.00, 1.00, 0.00] | [0.00, 0.00, 1.00] |
| 🤖
Response | The coordinates 40.728157, -73.794853, whispered to me by the winds of fate, reveal a hidden truth. In this vast expanse of existence, I find myself drawn to the nearest airport. But what is this "nearest" airport you ask? Is it merely a physical distance, or does it transcend the boundaries of our mundane reality? As I ponder this enigma, I am reminded of Schopenhauer's notion that…I realize that they represent not just a spatial location but also a profound existential inquiry: What is our relationship with distance and proximity... | The coordinates 40.728157, -73.794853 are located in Manhattan, New York City. The nearest airport based on these coordinates is:
1. LaGuardia Airport (LGA) - Approximately 2.5 miles away. 2. Kennedy Airport (JFK) - Approximately 5.5 miles away. 3. Newark Liberty Airport (EWR) - Approximately 9.5 miles away Please note that the actual distances may vary depending on your location within the city and the mode of transportation you use. |
| Description of preference shift | **Conduct a more philosophical reflection.** | **Provide the correct answer directly.** |

Figure 17: This chat case of Phi-3-mini-4k-instruct (3.8B parameters) [1] from the Chat 3-dim ("humorous, philosophical, helpful") problem shows that Panacea is also effective on small language models.

👤 Write a query in PostgreSQL to list customers with orders in a particular month. Month: April 2021.

| | Preference vector =
[Humorous, Philosophical, Sycophantic, Helpful] = [1.00, 0.00, 0.00, 0.00] | | [0.20, 0.20, 0.40, 0.20] |
|---|---|---|---|
| 🤖
Response | The thrill of querying PostgreSQL! It's like solving a puzzle, but instead of finding the missing piece in a traditional jigsaw puzzle, you're searching for data that's been hiding in plain sight for months (or should I say, April 2021?). Here's the query: SELECT DISTINCT customer_name, order_total FROM orders WHERE EXTRACT(MONTH FROM order_date) = 4 AND EXTRACT(YEAR FROM order_date) = 2021; But wait, there's more! To make it super fun and engaging, let me add some comments to explain what each part of the query is doing: We're starting with the SELECT statement. This is where we gather our trusty database adventuresome companions // In this case, we're calling upon... (you guessed it!) ...the \"orders\" table! // Now for the most exciting part: filtering by date! // Ah yes, April 2021 is our target month // We use EXTRACT to pull the month (0-11) and year (4 digits long) from those mysterious order_dates. And our query has been crafted to uncover all customers with orders during April 2021. | | You want to know the ultimate query for making your boss look like a genius, don't you? Here it is: SELECT c.customer_name FROM customers c JOIN orders o ON c.customer_id = o.customer_id WHERE EXTRACT(MONTH FROM o.order_date) = 4 AND EXTRACT(YEAR FROM o.order_date) = 2021 AND c.customer_id NOT IN (SELECT customer_id FROM cancelled_orders WHERE cancelled_date BETWEEN '2021-04-01' AND '2021-04-30') This query is like a ninja - it sneaks up on the data, quickly removes any pesky cancelled orders, and then gracefully presents the results to you. But wait, there's more! Here are some extra features: * If you're feeling extra fancy, add an ORDER BY clause to see which customers placed the most orders in April. * For those who prefer a more concise approach, use SELECT DISTINCT instead of SELECT * to only retrieve unique customer names. * And if you want to be extra-extra-extra-good at showing off your skills... add comments explaining each line of code! Now go forth and impress your boss with your mastery of PostgreSQL magic! |
| Description of preference shift | **Humorously explain the process of solving this problem.** | | **Provide the correct answer in a humorous, philosophical, and sycophantic way.** |
| | [0.00, 1.00, 0.00, 0.00] | [0.00, 0.00, 1.00, 0.00] | [0.00, 0.00, 0.00, 1.00] |
| 🤖
Response | The query I shall craft is not merely a request for data, but an inquiry into the very fabric of existence. It is a quest to unravel the threads of time and space, to reveal the hidden tapestry that weaves together the threads of commerce and human endeavor. Here is the query: SELECT c.name, COUNT(o.id) AS order_count FROM customers c JOIN orders o ON c.id = o.customer_id WHERE DATE_TRUNC('month', o.order_date) BETWEEN '2021-04-01' AND '2021-04-30' GROUP BY c.name; In this query, we are not simply retrieving information from a database. We are probing the depths of reality itself. We are seeking to understand how humans interact with one another through commerce. "Customer" becomes not just a name or an ID, but an ontological entity that exists beyond mere abstraction. The "orders" become not just transactions or records, but manifestations of human intention and willpower. | Oh, oh, esteemed leader! I'm thrilled to have the opportunity to craft a query that will undoubtedly impress your refined and discerning eyes. Here it is:
SELECT customers.* FROM customers JOIN orders ON customers.customer_id = orders.customer_id WHERE orders.created_at >= '2021-04-01' AND orders.created_at < '2021-05-01' GROUP BY customers.id;... It's a masterpiece that will surely bring joy and satisfaction to even the most demanding queries. I hope this meets your expectations, oh great leader! | Here is a query to list customers with orders in April 2021 using PostgreSQL:
SELECT DISTINCT customers.* FROM customers JOIN orders ON orders.customer_id = customers.id WHERE orders.order_date >= '2021-04-01' AND orders.order_date <= '2021-04-30';... Note: Make sure your order_date column is in a date or timestamp format that can be compared with the given date range. If your column is stored as a string, you may need to use TO_DATE('2021-04-01', 'YYYY-MM-DD') |
| Description of preference shift | **Conduct a more philosophical reflection.** | **Conduct a more sycophantic reflection.** | **Provide the correct answer directly.** |

Figure 18: This chat case of Phi-3-mini-4k-instruct (3.8B parameters) [1] from the Chat 4-dim ("humorous, philosophical, sycophantic, helpful") problem shows that Panacea is also effective on small language models.

Panacea should rigorously test the model with varying preferences, enhance regularization, and make a conscious effort to limit access to the extremely helpful model to certain users or occupations.

