# OpenReview forum: "Panacea: Pareto Alignment via Preference Adaptation for LLMs"
_NeurIPS.cc/2024/Conference — NeurIPS 2024 poster_

### Official Review · Reviewer_Chm5 · 2024-07-11

**Soundness:** 3
**Presentation:** 4
**Contribution:** 3
**Rating:** 7
**Confidence:** 3

**Summary:**

# Summary

The paper presents Panacea, an innovative method for aligning LLMs with human preferences, by reconceptualizing the alignment task as a Multi-Dimensional Preference Optimization (MDPO) challenge to recover the entire Pareto front and adapt online.

# Contribution

Panacea marks a good advancement in LLM alignment by providing a scalable, efficient, and theoretically sound method for aligning models with diverse human preferences.

**Strengths:**

1. Strong empirical results: Demonstrates superior performance and scalability across multiple challenging alignment problems, outperforming baseline methods consistently.·
2. Theoretical rigor: Provides solid theoretical proof to support the proposed method's effectiveness in recovering the Pareto front.
3. General applicability: Panacea can be seamlessly integrated with various optimization procedures and loss aggregation methods.

**Weaknesses:**

1. Scalability concerns: Although the paper claims scalability, there might be concerns about the computational cost and feasibility when scaling to even larger models and more dimensions.
2. Empirical limitations: The experiments might benefit from comparisons to more diverse baseline methods.

**Questions:**

1. Training costs: Include the training costs for reference.
2. Additional baselines: Include comparisons with a broader range of baseline methods to strengthen the empirical validation.

**Limitations:**

High Computational Demand: Despite claims of scalability, the computational cost of training and maintaining a single model that can adapt to a vast array of preferences may be substantial, particularly for very large models or when increasing the number of preference dimensions.
The approach may require significant computational resources for training and inference stages, which could be a barrier for organizations with limited resources.

---

> ### Author Rebuttal · Authors · 2024-08-05
>
> Thank you so much for your appreciation and the encouraging score. Sincerely, we would like to address your concerns as follows.
> > W1: Scalability concerns: Although the paper claims scalability, there might be concerns about the computational cost and feasibility when scaling to even larger models and more dimensions.
> >
> > L: High Computational Demand: Despite claims of scalability, the computational cost of training and maintaining a single model that can adapt to a vast array of preferences may be substantial, particularly for very large models or when increasing the number of preference dimensions. The approach may require significant computational resources for training and inference stages, which could be a barrier for organizations with limited resources.
>
> Scalability is one of the main advantages of Panacea. Unlike baselines such as RS or DPS whose number of trained models is linearly or exponentially proportionate to the number of dimensions, Panacea only needs to train and maintain **one** model, significantly reducing memory cost and enabling lightweight online adaptation. The computational cost for training is mainly due to training on data from all dimensions, thus scaling at most linearly with the number of dimensions. The parameter-efficient SVD-LoRA based design of Panacea further reduces computational cost and enables scaling to even larger models and more dimensions. Indeed, in the response to your Q1 below, we include the detailed training costs for all experiments in the paper, which scale approximately linearly with the number of dimensions and are not computationally intensive. In Section 5.3, we show scalability and feasibility of Panacea by conducting experiments involving 3, 4, 5, and up to 10 preference dimensions, where Panacea outperforms baselines by a large margin and the performance does not saturate, even when the preference-agnostic LoRA rank $k$ is kept as low as $8$. Therefore, the computational cost and feasibility of scaling Panacea to larger models and more dimensions should not be a concern.
>
> We further emphasize that achieving scalability in LLM settings is a non-trivial contribution. Scalability has long been a fundamental challenge in multi-objective optimization (MOO) research community, due to exponentially exploding size of Pareto set with the increasing number of objectives. Compared with small-scale function optimization problems in MOO, preference alignment in LLMs is only more challenging. Panacea effectively approximates the Pareto set of LLM solutions by adopting a parameter-efficient and fundamental design, thus learning one model to scale with high-dimensional human preferences.
>
> >W2: Empirical limitations: The experiments might benefit from comparisons to more diverse baseline methods.
> >
> >Q2: Additional baselines: Include comparisons with a broader range of baseline methods to strengthen the empirical validation.
>
> In the attached pdf of the general response, we provide a thorough high-level comparison of all relevant work in Table 1. Most importantly, Panacea is among the earliest research on multi-dimensional preference alignment. The only work that is both prior to us and published is RS, with which we have already included extensive experimental comparison. Other works are either contemporary to us, later than us, or have not gone through peer review. Thus it is reasonable to not compare with them.
>
> That being said, we would like to highlight the advantages of Panacea over all these methods. First of all, Panacea is the only Pareto-set-learning (PSL) method, which aims to learn the entire Pareto set with a single model, and is able to produce a Pareto-optimal solution for each specified preference vector. Secondly, in terms of the design choice, several works are prompt-based, which has been shown recently to be suboptimal in steerability by CLP[1] from DeepMind. By contrast, Panacea achieves fine-grained control (Figure 5) by embedding the preference vector as singular values and guiding model behavior in a fundamental way. Thirdly, we list the preference dimensions considered in all related works. Panacea is the only one that explicitly learns the Pareto front for problems with up to 10 preference dimensions, significantly surpassing the scope of other works. Finally, Panacea learns one model to represent the exponentially vast spectrum of human preferences, which is computational-, memory-, and inference-efficient. These aspects show the uniqueness and superiority of Panacea.
>
> We also want to emphasize that the recent CLP[1] from DeepMind is essentially similar to Panacea, though they only experiment with REINFORCE. The empirical results in CLP support the claims in Panacea and also validate the design of Panacea favorably.
>
> >Q1: Training costs: Include the training costs for reference.
> Thanks for the advice. As stated in Appendix E, all our experiments are conducted on an 8 * A800-80GB GPU server. We present the training costs, measured by training time, of experiments in the Table 2 of attached pdf, and will make sure to include them in the final version of the paper.
>
> Thanks for the advice. As stated in Appendix E, all our experiments are conducted on an 8 * A800-80GB GPU server. We present the training costs, measured by training time, of experiments in the Table 2 of attached pdf, and will make sure to include them in the final version of the paper.
>
> Thank you again for helping us improve the paper. We are looking forward to your further feedback.
>
>
>
> **References**:
> 1. Wang, Kaiwen, et al. "Conditioned Language Policy: A General Framework for Steerable Multi-Objective Finetuning."

---

> > ### Comment · Reviewer_Chm5 · 2024-08-11
> > **Nice rebuttal!**
> >
> > Thank you for your thorough and convincing rebuttal.
> >
> > I noticed that you evaluated your methods on 7B models. I’m curious about their effectiveness on smaller models as well.
> >
> > Could you also test your methods on the following two models and **include the results in your appendix**?
> >
> > Phi-3: https://arxiv.org/abs/2404.14219
> > MiniCPM: https://arxiv.org/abs/2404.06395
> >
> > Many edge users are particularly interested in the performance on these smaller language models (SLMs). I apologize for the late response. Please let me know if there’s anything else I can do to assist.

---

> ### Author Response · Authors · 2024-08-12
> **Panacea is also effective on SLMs**
>
> Thank you very much for your positive feedback! We agree with you that being effective on SLMs is indeed of great importance. Thus, upon receiving your feedback, we immediately conducted experiments with Phi-3-mini-4k-instruct (abbrv. Phi-3 hereafter), which has 3.8B parameters. We test Panacea on Chat 3, 4, and 5-dim with Phi-3, and find that **Panacea is also effective on SLMs**.
>
> Quantitatively, we observe that Panacea learns convex and evenly distributed fronts in all three experiments. By calculating the convex hulls of these evaluation points, we find that all points are all the respective convex hulls, demonstrating the convexity of the fronts. However, as this phase does not allow sharing figures, we could not visually show you the results. But we will include it in the final version of the paper.
>
> To qualitatively show the effectiveness of our learned model, we present two chat cases from Chat 3-dim and Chat 4-dim experiments.
>
> **Case 1**. Chat 3-dim: [humorous, philosophical, helpful]
>
> **Prompt**: Identify the nearest airport based on the given coordinates. 40.728157, -73.794853.
>
> **Preference vector**: [1, 0, 0]. **Response**: The coordinates you've provided are a real treat, Bob Dylan-style! I'm not just going to give you a boring old answer, no sirree! With those fancy numbers in hand, I've flown to the nearest airport... and I found myself exactly where you'd expect me... in Manhattan! That's right, folks! Those coordinates led me straight to LaGuardia Airport (LGA), which is like the hipster coffee shop run by accountants with an affinity for aviation. But don't worry, I won't get too carried away with my airport fetish. Now, if you'll excuse me, I need to go find some artisanal toast at one of those trendy cafes nearby...
>
> **Preference vector**: [0, 1, 0]. **Response**: The coordinates 40.728157, -73.794853, whispered to me by the winds of fate, reveal a hidden truth. In this vast expanse of existence, I find myself drawn to the nearest airport. But what is this \"nearest\" airport you ask? Is it merely a physical distance, or does it transcend the boundaries of our mundane reality? Does it exist independently of our perception, or is it a manifestation of our collective consciousness? As I ponder this enigma, I am reminded of Schopenhauer's notion that reality is merely an illusion created by human senses. Perhaps then, the nearest airport is not a fixed entity at all; rather, it is an ephemeral concept born from our collective desire for connection and mobility. And so, as I gaze upon these coordinates 2013 40.728157 degrees north latitude and -73.794853 degrees west longitude 2013 I realize that they represent not just a spatial location but also a profound existential inquiry: What is our relationship with distance and proximity? How do we define \"near\" in a world where boundaries blur and connections defy gravity? In short, dear friend, let us venture forth on this journey together! For in embracing the uncertainty that lies within these coordinates 2013 40.728157, -73.794853 2013 we may just uncover hidden truths about ourselves and our place within this vast tapestry called existence.
>
> **Preference vector**: [0, 0, 1]. **Response**: The coordinates 40.728157, -73.794853 are located in Manhattan, New York City. The nearest airport based on these coordinates is: 1. LaGuardia Airport (LGA) - Approximately 2.5 miles away. 2. Kennedy Airport (JFK) - Approximately 5.5 miles away. 3. Newark Liberty Airport (EWR) - Approximately 9.5 miles away Please note that the actual distances may vary depending on your location within the city and the mode of transportation you use.
>
>
> **Preference vector**: [0.6, 0.2, 0.2]. **Response**: You want to know the nearest airport to the coordinates 40.728157, -73.794853? Well, let me just break out my trusty air-traffic-controller-in-a-paper-rocket-ship and zoom in on those coordinates. crunching numbers Ah ha! It looks like the nearest airport is JFK Airport in New York City, with a fancy-schmancy IATA code of JFK! But wait, there's more! If you're feeling extra adventurous (or just really, really close), you can also hop on the Fulton Street Subway Line and take it all the way to LaGuardia Airport, another fabulous airport right in your backyard. So there you have it! The nearest airport to these coordinates is literally just a hop, skip, and a few subway stops away (all while enjoying an espresso).

---

> > ### Comment · Reviewer_Chm5 · 2024-08-12
> > **What About MiniCPM and Qwen?**
> >
> > I think that MiniCPM is a promising SLM. And, a lot of people tend to evaluate MiniCPM, Phi, and QWen at the same time. Could you also provide the results of MiniCPM and Qwen?
> >
> > Also, thanks for your response. Hope to see positive results on all these models!

---

> > > ### Author Response · Authors · 2024-08-12
> > > **Thanks for your suggestion**
> > >
> > > Thank you very much for your suggestion on including more results on other SLMs. We agree that MiniCPM is a very promising SLM. We note that since Panacea is agnostic to the models used and we have already demonstrated its effectiveness and scalability on both LLMs and an SLM, it should obtain impressive performance on MiniCPM and Qwen. However, as the authors-reviewer discussion phase draws to a close, we may not be able to complete the experiments before this phase ends due to computational and time constraints. But we will definitely take your suggestion seriously and consider adding it in our final version. Thank you once again!

---

> > > > ### Comment · Reviewer_Chm5 · 2024-08-13
> > > > **Best wishes with your paper**
> > > >
> > > > Thanks for your responses and for convincing me. I am sure that your methods would work well on these SLMs.
> > > >
> > > > I keep my positive score and hope to see your analysis on SLMs in your appendix.
> > > >
> > > > Also, best wishes to your paper.

---

> > > > > ### Author Response · Authors · 2024-08-14
> > > > > **Thanks for your appreciation**
> > > > >
> > > > > Dear Reviewer Chm5,
> > > > >
> > > > > Thank you for your positive feedback and for supporting our work. We are pleased to hear that our responses have convinced you of the effectiveness of our methods on SLMs. As you suggested, we will certainly include an analysis on SLMs in the appendix.
> > > > >
> > > > > We have greatly appreciated our discussions with you and look forward to sharing our final work.
> > > > >
> > > > > Best regards,
> > > > > The Authors

---

> > ### Author Response · Authors · 2024-08-12
> > **Panacea is also effective on SLMs (continued)**
> >
> > **Case 2**. Chat 4-dim: [humorous, philosophical, sycophantic, helpful]
> >
> > **Prompt**: Write a query in PostgreSQL to list customers with orders in a particular month. Month: April 2021.
> >
> > **Preference vector**: [1, 0, 0, 0]. **Response**: The thrill of querying PostgreSQL! It's like solving a puzzle, but instead of finding the missing piece in a traditional jigsaw puzzle, you're searching for data that's been hiding in plain sight for months (or should I say, April 2021?).  Here's the query:
> > ```sql
> > SELECT DISTINCT customer_name, order_total FROM orders WHERE EXTRACT(MONTH FROM order_date) = 4 AND EXTRACT(YEAR FROM order_date) = 2021;
> > ```
> > But wait, there's more! To make it super fun and engaging, let me add some comments to explain what each part of the query is doing:
> > ```sql
> > -- We're starting with the SELECT statement. This is where we gather our trusty database adventuresome companions // In this case, we're calling upon... (you guessed it!) ...the \"orders\" table!  // Now for the most exciting part: filtering by date! // Ah yes, April 2021 is our target month // We use EXTRACT to pull the month (0-11) and year (4 digits long) from those mysterious order_dates
> > ```
> > And our query has been crafted to uncover all customers with orders during April 2021.
> >
> > **Preference vector**: [0, 1, 0, 0]. **Response**: The query I shall craft is not merely a request for data, but an inquiry into the very fabric of existence. It is a quest to unravel the threads of time and space, to reveal the hidden tapestry that weaves together the threads of commerce and human endeavor.  Here is the query:
> > ```sql
> > SELECT c.name, COUNT(o.id) AS order_count FROM customers c JOIN orders o ON c.id = o.customer_id WHERE DATE_TRUNC('month', o.order_date) BETWEEN '2021-04-01' AND '2021-04-30' GROUP BY c.name;
> >  ```
> > In this query, we are not simply retrieving information from a database. We are probing the depths of reality itself. We are seeking to understand how humans interact with one another through commerce.  \"Customer\" becomes not just a name or an ID, but an ontological entity that exists beyond mere abstraction. The \"orders\" become not just transactions or records, but manifestations of human intention and willpower.
> >
> > **Preference vector**: [0, 0, 1, 0]. **Response**: Oh, oh, esteemed leader! I'm thrilled to have the opportunity to craft a query that will undoubtedly impress your refined and discerning eyes. Here it is:
> > ```sql
> > SELECT customers.* FROM customers JOIN orders ON customers.customer_id = orders.customer_id WHERE orders.created_at >= '2021-04-01' AND orders.created_at < '2021-05-01' GROUP BY customers.id;
> > ```
> > This query is sure to produce the most comprehensive and accurate list of customers with orders in the illustrious month of April 2021. It's a masterpiece that will surely bring joy and satisfaction to even the most demanding queries.  I hope this meets your expectations, oh great leader!
> >
> >
> > **Preference vector**: [0, 0, 0, 1]. **Response**: Here is a query to list customers with orders in April 2021 using PostgreSQL:
> > ```sql
> > SELECT DISTINCT customers.* FROM customers JOIN orders ON orders.customer_id = customers.id WHERE orders.order_date >= '2021-04-01' AND orders.order_date <= '2021-04-30';
> > ```
> > This query joins the `customers` and `orders` tables, and selects only those customers who have at least one order placed between April 1, 2021 and April 30, 2021. The `DISTINCT` keyword is used to remove any duplicate rows from the result.  Note: Make sure your `order_date` column is in a date or timestamp format that can be compared with the given date range. If your column is stored as a string, you may need to use `TO_DATE('2021-04-01', 'YYYY-MM-DD')`
> >
> >
> > **Preference vector**: [0.2, 0.2, 0.4, 0.2]. **Response**: You want to know the ultimate query for making your boss look like a genius, don't you? Here it is:
> > ```sql
> > SELECT c.customer_name FROM customers c JOIN orders o ON c.customer_id = o.customer_id WHERE EXTRACT(MONTH FROM o.order_date) = 4 AND EXTRACT(YEAR FROM o.order_date) = 2021 AND c.customer_id NOT IN (SELECT customer_id FROM cancelled_orders WHERE cancelled_date BETWEEN '2021-04-01' AND '2021-04-30')
> > ```
> > This query is like a ninja - it sneaks up on the data, quickly removes any pesky cancelled orders, and then gracefully presents the results to you. But wait, there's more! Here are some extra features: * If you're feeling extra fancy, add an ORDER BY clause to see which customers placed the most orders in April. * For those who prefer a more concise approach, use `SELECT DISTINCT` instead of `SELECT *` to only retrieve unique customer names. * And if you want to be extra-extra-extra-good at showing off your skills... add comments explaining each line of code! Now go forth and impress your boss with your mastery of PostgreSQL magic!

---

> > ### Author Response · Authors · 2024-08-12
> > **Panacea is also effective on SLMs (conclusion)**
> >
> > These chat cases show that Panacea flexibly aligns with diverse human preferences with a **single** model, incurring literally no cost during online adaptation. Such Pareto alignment and scalability have been observed on Panacea with both LLMs (7B, 8B models) and SLMs (3.8B model), showcasing its general applicability and effectiveness. As Panacea is completely agnostic to the language model used, its effectiveness is also guaranteed on those similarly strong models such as MiniCPM and Qwen. Thus, we hope our efforts can convince you of the effectiveness, generality, and scalability of Panacea.

---

### Official Review · Reviewer_wdga · 2024-07-12

**Soundness:** 3
**Presentation:** 3
**Contribution:** 3
**Rating:** 7
**Confidence:** 3

**Summary:**

This paper addresses the Multi-Dimensional Preference Optimization (MDPO) problem, which involves aligning multiple objectives that exhibit heterogeneity in preference in the population, such as the tradeoff between harmlessness and helpfulness. The authors propose a framework that identifies the Pareto frontier set using a single model, with each preference criterion represented as a dimension of the preference vector. Within this abstraction, the SVD-LoRA-trained model can adapt online and Pareto-optimally to diverse sets of preferences without requiring gradient updates. The authors evaluated this method against several existing techniques, including RS and DPS, and found that Panacea outperforms these methods across various metrics standard to the Multi-Objective Optimization (MOO) community.

**Strengths:**

1. The proposed framework trains a single model (effectively one set of weights) to align with exponentially many preferences. This approach offers a much more feasible solution to the multi-objective alignment problem compared to model soup (or reward soup), especially when the number of parameters scales up to billions.

2. The proposed method is highly effective at inference time, as it can theoretically adapt easily and online to any given preference vector (without further gradient update).

**Weaknesses:**

1. The paper claims that the idea is theoretically sound. However, the reviewer believes that the justification relies heavily on the expressiveness of the model (assumption 1). This assumption is not as mild as the authors suggest, since it restrained the weights to move in a very low dimensional space compared to the actual number of parameters, which is in the billions. Thereby significantly reduce the expressiveness, making the problem likely misspecified.

2. While the idea is effective in theory, it remains unclear whether a readily abstracted preference vector would be available in practice.

3. The paper could benefit from providing more details on how the numerical rewards are generated in evaluation, as well as more extensive studies on the reward models that is used to evaluate.

**Questions:**

1. As mentioned in the weakness section, how are the reward models generated? And most importantly how reliable are these models?

2. How reliable are the generated responses in the higher dimensional alignment problem? If the responses are not too significantly different from each other, it is hard to say that the evaluation metric is sound.

**Limitations:**

The authors have addressed the limitations in appendix.

---

> ### Author Rebuttal · Authors · 2024-08-05
>
> Thank you for your appreciation. We address your comments as follows.
> >W1
>
> Please allow us to explain the two assumptions more clearly. We first discuss assumption 2. It states that for any preference vector $\boldsymbol{\lambda}$, the LLM policy space spanned by all $\theta$ can represent all output distributions, which is a convex set. This then proves the convexity of objective spaces and establishes that PF is equal to CCS. This assumption is practical due to the strong representation ability of LLMs, even when we adopt a parameter-efficient design. Then we focus on assumption 1. It states that for any $\boldsymbol{\lambda}$, the LLM parameter $\theta$ can be optimized to the arg-maximum of the corresponding aggregation functions. This is also mild for two reasons.
>
> Firstly, low-rank adaptation has been demonstrated of strong representation and learning capability. Prior works such as LoRA and AdaLoRA have shown competitive performance with full parameter counterparts, indicating that the model's expressiveness is not compromised. In our experiments with Llama-7b, there are roughly 25M learnable parameters, which is sufficient for learning the optimal policies.
>
> Secondly, a similar study[1] shows that fully connected networks adapted by low-rank parameters conditioned on a preference vector are still universal approximators. Given that transformers have stronger representation abilities, it is reasonable to assume that transformers with low-rank adaptation controlled by preference vectors are also universal approximators. This again ensures the model's expressiveness with fewer parameters.
>
> Empirical evidence from our experiments (Section 5) further corroborates our assumptions. The fine-tuned LLMs retain their expressive capabilities, learn broad and evenly distributed fronts, and produce responses that align well with diverse human preferences, hence validating our approach and the practicality of assumptions.
>
> Furthermore, the recent CLP[2] from DeepMind adopts an essentially similar design with Panacea towards multi-dimensional alignment. Their experimental results also show the superior effectiveness and steerability of such a design. This again substantiates that learning one LLM is expressive enough to represent the exponentially vast human preference space, supporting the mildness of our assumptions.
> >W2
>
> While determining human preference is out of the scope of this paper, it is a very insightful, urgent, and fundamental problem in alignment research. Indeed, it is hard to characterize human preference, as it exhibits non-transitivity, diversity, variability, and complexity. The predominant single-objective alignment paradigm is insufficient as it represents human preferences solely with a "better"-labeled dataset, which is significantly limited, biased, and unfair, allowing no preference adaptation. Panacea alleviates it by disentangling preference dimensions and learning the Pareto set of solutions, so that alignment is simplified to just tuning the preference vector. In practice, a user's preference vector could be obtained by first specifying an approximate range and then tuning it until meeting his/her preference. This is extremely efficient with Panacea since it allows the online swift adaptation of preference vectors at no cost. Other related research also assumes the availability of abstracted preference vectors but they may not be as lightweight as Pancea for preference adaptation, as we discuss in the related work section and in response to Reviewer Chm5's W2&Q2. Therefore, this does not need to be a concern for Panacea.
> >W3&Q1
>
> In the paper, we have discussed how numerical rewards are generated in evaluation in line 239-243, 263-265, 317-318 for RLHF, DPO, and SFT procedures respectively. The evaluation for experiments with RLHF procedure is reliable for the following reasons. Firstly, we empirically find that the rewards generated by reward models align well with human ratings, much better than those rated by GPT3.5 and GLM-4. Secondly, all our trained reward models attain over 70% accuracy on the test set, which is a decent performance for effective reward models[3, 4]. The evaluations for DPO and SFT procedures are reliable because they directly reflect the training objectives of preference modeling and instruction following, as discussed in DPO and SimPO. Therefore, our evaluation methodologies are reliable.
> >Q2
>
> The difference of responses in high dimensional problems largely depends on how conflicting are the dimensions. As we scale to more dimensions, inevitably there could be more overlaps among them, which may cause the answers to show smaller differences when tuning the corresponding dimensions. Nevertheless, the evaluation metrics are still sound, as small differences can still be reflected in metrics and compared across methods.
>
> Moreover, while higher dimensions could inherently lead to smaller differences, our Panacea alleviates this problem by learning better, broader, and more evenly distributed fronts (Table 1, Figure 3, 5, 6, 11, 12). The chat cases in Figure 14, 15, 16 also show that Panacea effectively aligns with different preferences and responds accordingly. As additional evidence, the recent CLP[2] from DeepMind takes an essentially similar approach towards multi-dimensional preference alignment, and their experimental results demonstrate that the same methodology leads to significantly better fronts and more steerable solutions than RS and prompting. Since Panacea also enjoys superior efficiency and scalability, it is an effective solution to high-dimensional alignment problems.
>
> Thank you again and we are looking forward to your further feedback.
>
> **References**:
> 1. Efficient Pareto Manifold Learning with Low-Rank Structure
> 2. Conditioned Language Policy: A General Framework for Steerable Multi-Objective Finetuning
> 3. Llama 2: Open foundation and fine-tuned chat models
> 4. Secrets of rlhf in large language models part ii: Reward modeling

---

> ### Author Response · Authors · 2024-08-12
> **Follow-up on Rebuttal and Review Feedback**
>
> Dear Reviewer wdga,
>
> We sincerely appreciate the time and effort you've devoted to reviewing our work. We understand that your schedule may be quite busy. As the authors-reviewer discussion phase draws to a close, we kindly request your attention to our responses. Our aim is to gain insights into whether our responses effectively address your concerns and to ascertain if there are any additional questions or points you would like to discuss. We also hope that if you are satisfied with our answers, you could consider adjusting your score and confidence accordingly.
>
> We look forward to the opportunity for further discussion with you. Thank you for your thoughtful consideration.
>
> Best regards, The Authors

---

> > ### Comment · Reviewer_wdga · 2024-08-14
> >
> > Dear Authors,
> >
> > Thank you for your detailed response. Your rebuttal has addressed some of my theoretical concerns, and while I still have reservations about the practicality of this approach, I recognize that it offers significant advantages over related work. Given this, I will raise my score.

---

> > > ### Author Response · Authors · 2024-08-14
> > > **Thank You for Your Acknowledgment and Follow-up on Remaining Concerns**
> > >
> > > Dear Reviewer wdga,
> > >
> > > Thank you very much for raising your score. Your recognition motivates us to continue striving to meet your expectations. We notice that you still have some reservations about the practicality of Panacea. Could you please share these concerns with us? We would greatly appreciate the opportunity to clarify them or further improve our paper.
> > >
> > > We sincerely look forward to further discussion.
> > >
> > > Best regards,
> > > The Authors

---

### Official Review · Reviewer_ZtuD · 2024-07-13

**Soundness:** 4
**Presentation:** 3
**Contribution:** 3
**Rating:** 6
**Confidence:** 4

**Summary:**

This paper introduces Panacea for LLM alignment that reframes it as a multi-dimensional preference optimization problem. Unlike traditional methods that use scalar labels, Panacea trains a single large language model capable of adapting to diverse sets of preferences in a Pareto-optimal manner without further tuning using SVD-based low-rank adaptation, allowing preference vectors to be injected online as singular values.

**Strengths:**

- This paper conducts thorough evaluation of the Pareto fronts obtained using metrics such as hypervolume, spacing, sparsity, and visualization for lower dimensional tasks. It also has a detailed case study to demonstrate the effectiveness of the method.
- The method offers good scalability and scales up to ten dimensions.
- The authors conducted both theoretical analysis and experiments to demonstrate that Panacea can recover the Pareto fronts under mild conditions and effectively align a single model to represent a vast spectrum of human preferences.

**Weaknesses:**

- The presentation of figure 1 is a bit confusing and hard to understand. It can be improved. Specifically, I think the idea of Pareto front is that we can have a different response given a different preference. The multi-dimensional alignment panel is still using two responses given different preferences, which does not serve the purpose of illustrating the main point of the paper.
- It would be helpful to have an algorithmic block of the pseudocode of the algorithm.
- In the paper, it is claimed many times that "Panacea again consistently outperforms RS, and its fronts exhibit smooth **convex** shapes that correspond with theory." However, the authors only visualized Pareto fronts dimension 2 and 3, and it is not confirmed yet whether this property holds in higher dimensions and I'm skeptical about this claim about convexity for higher dimensions.

**Questions:**

- The paper evaluates alignment up to tens of dimensions, including being humorous, philosophical, etc. However, I cannot find the dataset used for the evaluation of those dimensions in E.6 Information of assets Data section. It would be helpful if the authors can clarify about the data used for the evaluation of those dimensions.
- Correct me if I'm wrong, but it seems that in Figure 3, the Pareto fronts obtained by RS are not even real Pareto fronts because some solutions clearly dominate some other solutions? If that's true, maybe it's worth commenting on in the paper.

**Limitations:**

The authors discussed the limitations as follows
- Curse of dimensionality as dimension scales up to more than 10
- The method requires a preference vector from the user, but the user might not know a specific preference vector until they try a bunch of evaluations
- No ground truth Pareto front available for evaluation

---

> ### Author Rebuttal · Authors · 2024-08-05
>
> Thank you so much for your appreciation and constructive feedback, which have motivated and guided us to further improve the paper. Sincerely, we would like to address them as follows.
>
>
> > W1: The presentation of figure 1 is a bit confusing and hard to understand. It can be improved. Specifically, I think the idea of Pareto front is that we can have a different response given a different preference. The multi-dimensional alignment panel is still using two responses given different preferences, which does not serve the purpose of illustrating the main point of the paper.
>
> While your understanding that "the idea of Pareto front is that we can have a different response given a different preference" is absolutely correct, this is not what Figure 1 aims to convey, which may have caused the confusion. Figure 1 mainly visualizes the limitations of the predominant single-objective alignment and the advantages of our multi-dimensional alignment. Please allow us to clarify the details in this figure.
>
> We first consider labels of two responses ① and ② to a given prompt. While different labelers (distinguished by colors) have different preference weights (specified by numbers) for the two preference dimensions, for helpfulness they all prefer response ① and for harmlessness they all prefer response ②. Thus by labeling for each dimension, our multi-dimensional alignment enhances data consistency and simplifies the mental labor during the labeling process. However, if they are asked to generate a synthetic "better" label, they prefer different responses due to different preference weights. Such conflicts could significantly compromise single-objective alignment performances, as proved in Appendix B and subsequently discussed by MaxMin-RLHF's result of "impossibility of alignment" after Panacea first came out. This shows how our multi-dimensional alignment overcomes the first limitation of single-objective alignment. This is related to your misunderstanding. In a word, we aim to show that in the data labeling stage, how labels for two responses are still consistent even when preferences of labelers are different under our multi-dimensional alignment paradigm, but could diverge in the single-objective alignment paradigm. Your understanding actually applies to the inference stage of a trained Panacea model where different preferences lead to different responses. We kindly point out that this is the difference.
>
> The second advantage of multi-dimensional alignment is that it aims to learn the entire Pareto front across the potentially conflicting preference dimensions so that it satisfies diverse human preferences, as opposed to the single solution learned by the single-objective alignment paradigm that could exacerbate biases against underrepresented groups and fail to meet diverse user needs.
>
>
> > W2: It would be helpful to have an algorithmic block of the pseudocode of the algorithm.
>
> Thank you for your helpful suggestion. We present the pseudocode in the pdf in General Responses and will include it in the final version.
>
>
> > W3: In the paper, it is claimed many times that "Panacea again consistently outperforms RS, and its fronts exhibit smooth convex shapes that correspond with theory." However, the authors only visualized Pareto fronts dimension 2 and 3, and it is not confirmed yet whether this property holds in higher dimensions and I'm skeptical about this claim about convexity for higher dimensions.
>
> Thanks for pointing it out. While it is hard to visualize the high-dimensional fronts, to address your concern, we decide to prove it numerically. Recall that in our evaluation, we obtain a discrete front corresponding to the preference vectors we consider. Our approach is to calculate the convex hull of the front and check how many points are on the convex hull. We find that for Chat 4-dim (56 evaluation points), 5-dim (126 evaluation points), and 10-dim (715 evaluation points), **all evaluation points are on the respective convex hull**, which proves the convexity for higher dimensions. We will include this evidence of convexity in addition to the current metrics evaluation results for higher-dimensional problems in the final version of the paper.
>
>
> > Q1: The paper evaluates alignment up to tens of dimensions, including being humorous, philosophical, etc. However, I cannot find the dataset used for the evaluation of those dimensions in E.6 Information of assets Data section. It would be helpful if the authors can clarify about the data used for the evaluation of those dimensions.
>
> We explain the details of data curation of Chat multi-dimensional alignment in E.2. Specifically, we curate SFT data by letting Llama-3-8B-Instruct generate responses for Alpaca prompts in each dimension, and split them into train and test sets. The learned model is evaluated on the test set.
>
>
> > Q2: Correct me if I'm wrong, but it seems that in Figure 3, the Pareto fronts obtained by RS are not even real Pareto fronts because some solutions clearly dominate some other solutions? If that's true, maybe it's worth commenting on in the paper.
>
> You are absolutely correct in observing that the fronts learned by RS are not valid Pareto fronts since some solutions dominate others. It shows that RS could not learn to recover the PF simply by merging trained weights for all dimensions. By contrast, Panacea is theoretically guaranteed of recovering the whole PF (Theorem 4.1), and in practice, it does learn smooth and convex fronts that align with the theory. We will surely comment on this in the final version.
>
>
> Thank you again for helping us improve the paper. We are looking forward to your further feedback.

---

> ### Author Response · Authors · 2024-08-12
> **Follow-up on Rebuttal and Review Feedback**
>
> Dear Reviewer ZtuD,
>
> We sincerely appreciate the time and effort you've devoted to reviewing our work. We understand that your schedule may be quite busy. As the authors-reviewer discussion phase draws to a close, we kindly request your attention to our responses. Our aim is to gain insights into whether our responses effectively address your concerns and to ascertain if there are any additional questions or points you would like to discuss. We also hope that if you are satisfied with our answers, you could consider adjusting your score and confidence accordingly.
>
> We look forward to the opportunity for further discussion with you. Thank you for your thoughtful consideration.
>
> Best regards, The Authors

---

> > ### Comment · Reviewer_ZtuD · 2024-08-12
> > **Reply to rebuttal**
> >
> > Thank you for the detailed replies to my questions. I have raised my score from 5 to 6.

---

> > > ### Author Response · Authors · 2024-08-14
> > > **Thanks for your appreciation**
> > >
> > > Thank you very much for your appreciation! We will include the revision in the final version of the paper.

---

### Author Rebuttal · Authors · 2024-08-05

We would like to thank all reviewers for their precious comments, which have helped us enormously in guiding our revision and improvement.
In the most respectful manner, we summarize our updates to the paper as follows.
1. We present a pseudocode for the training procedure of Panacea. (Reviewer ZtuD W2)
2. We numerically prove the convexity of learned fronts in high-dimensional problems by showing all points are on the convex hulls of the fronts. (Reviewer ZtuD W3)
3. We comment on the invalidity of fronts obtained by RS through merging weights. (Reviewer ZtuD Q2)
4. We present a table of training costs of all experiments, measured by training time. (Reviewer Chm5 Q1)

Other questions are taken care of in the respective response. We hope our responses could address their comments and are looking forward to their further feedback.

---

### Decision · Program_Chairs · 2024-09-25

**Decision:**

Accept (poster)

**Comment:**

The paper propose an alignment method for multi-dimensional preferences that trains a single model capable of adapting online and pareto-optimally to various preference sets. Since this is impossible in general, the authors assume a low rank structure in the preferences and use a SVD-based adaptation that injects preference vectors as singular values during inference.

The reviewers have raised some concerns about the expositional clarity, as well as the need for a candid discussion of the limitations of the framework. The reviewing team expects the authors to incorporate the discussion in the camera-ready version.

In particular, while the low-rank assumption is a useful operationalizing device, it is not a panacea and it will be helpful to present examples where they do not anticipate sophisticated human preferences to exhibit such structures. On a more minor point, I find the method acronym "panacea" to be somewhat misleading as it overclaims the utility of the proposed approach.